# Geoheritage Management in Areas with Multicultural Interest Contexts

**Eva Pescatore, Mario Bentivenga** and **Salvatore Ivo Giano** *

Dipartimento di Scienze, University of Basilicata, Campus Macchia Romana Via Ateneo Lucano, 10, 85100 Potenza, Italy
* Correspondence: ivo.giano@unibas.it; Tel.: +39-09-7120-5842

**Abstract:** Sites of geo-cultural interest are often included in areas where multicultural contexts (geo and non geo) are present. Cultural heritage dissemination is sometimes mono-contextual, paying little attention to the possibility of inclusion in a wider multicultural context. When these different contexts are linkable to each other following a specific theme, multicultural heritage dissemination will be possible, and often the geo context can represent a fulcrum, a resilient tool in doing that. A portion of the Sinni river's catchment area (Basilicata region, Southern Italy) has been chosen to test and verify the multi-level/disciplinary approach applicability. The area is located on the southeastern edge of the Pliocene to Pleistocene Sant'Arcangelo basin in the Southern Apennines chain of Italy. Here, both basic observations on the physical geography landscape evolution and specialized observations on river dynamics and on the hydrographic network have been carried out. Educational routes will be proposed with different educational levels along a path that will include the San Giorgio Lucano hypogea. This paper represents the results of a qualitative study providing an overview of the possibility, in a multicultural context, about whether, when, and how the geo context may act as a link between the different disciplines and what is the best way to make it. A relational database, organized in contexts, areas, and themes, is planned at different levels of detail, and is currently being developed in order to make final products easily available. Each level will be provided with basic concepts, territorial contextualization, and of activities/itineraries. The goal is to provide a versatile tool that enhances the territorial multi-cultural heritage to reach a greater number of end users interested in both geo and non geo contexts.

**Keywords:** geoheritage management; area of multicultural interest (AMI); southern Italy

## 1. Introduction

In the last decades, several definitions, concepts, and study approaches concerning geodiversity, geological heritage, the geosites, and the geoconservation have characterized the scientific literature, as well as their dissemination and divulgation ([1–3] and references therein). The geoheritage represents a relevant amount of cultural and natural heritage. Usually, the sites of cultural interest are inserted in contexts characterized by geo and non-geo aspects (compound geo(morpho)site (*sensu* [4]). From a conceptual point of view, compound geo(morpho)sites can have different areal extensions ranging from a few to several km$^2$ (see examples in [4]). Where they are widely extended, it is possible to recognize the Areas of Multicultural Interest (AMI) containing several compound geo(morpho)sites, which can be related to each other through different themes or topics [5]. Most of the literature on heritage issues is specific and sectoral, dedicated to a single cultural context (historical, archaeological, geological, etc.). In a multicultural context, discriminating whether, when, and how the geo-context can act as a link between different disciplines and what is the best way to do so is a further way of addressing heritage issues. Geo aspects may represent a topic around which, and with which, prepare materials and contents for educational, informative, and tourist purposes. This multidisciplinary approach can also be used

for the production of materials and contents to support territorial planning actions aimed at territorial enhancing and protecting. In this paper, just aspects related to educational, informative, and tourist purposes are considered. So, the main end-users considered are represented by primary and secondary school students, university students, and tourists, for whom the contents have been prepared (concepts and vocabulary) according to the corresponding level considered.

Currently, the relationship between man and his environment is of particular interest, either in the basic cognitive aspects (knowledge of the environment in which we live and of its evolutionary dynamics) in the use of the natural resources (use/abuse) and in the protection of the natural environment (governance, protection, and maintenance). Man, during his evolutionary path, has always used what nature offered to him. Human/environment interactions have been the subject of reflections since the late 1800s [6,7] to continue over the years considering man as a Geomorphic Agent (capable of inducing substantial changes in the physical environment) whose effect on the physical environment has gradually increased exponentially with respect to its evolutionary and technological progress [8–11] and references therein. Notably, Stoppani (1873) [7], in defining man as a new element in nature, proclaims the Anthropozoic Era, during which "*man breaks down what nature has composed*" (Figure 1). In recent times, the term Anthropocene [12–16]; (http://quaternary.stratigraphy.org/working-groups/anthropocene/ (accessed on 1 May 2022)) referred to the geological period in which human action significantly affects the natural environment in all of its physical, chemical, and biological components, both locally and globally. "Anthropic affecting" that can be reflected on the Holocene stratigraphic record, overlapping the "natural" one linked to climatic/eustatic variations.

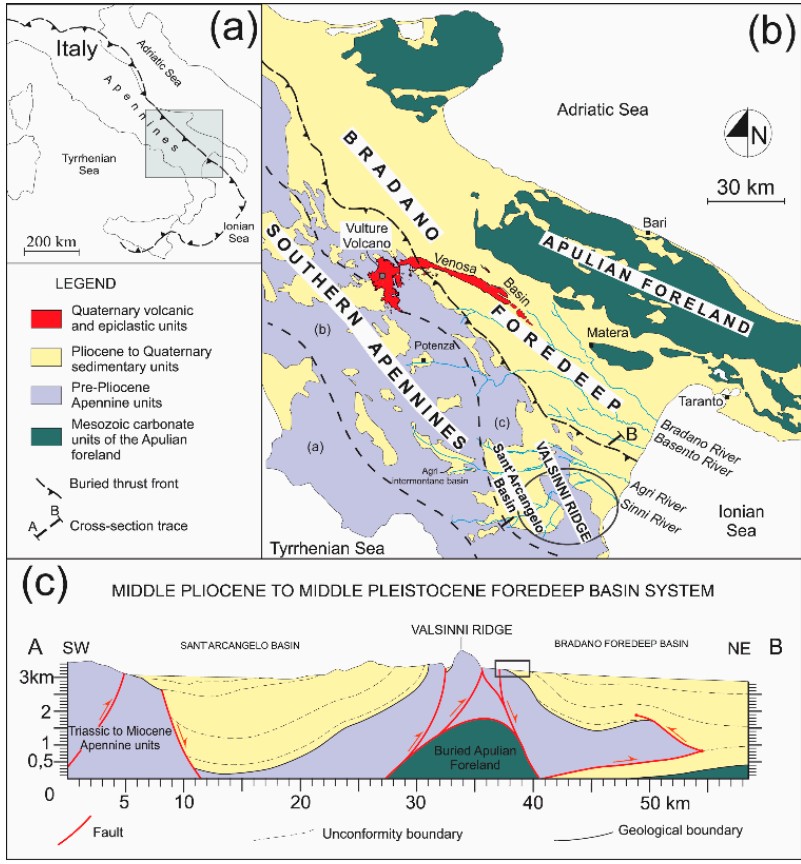

**Figure 1.** Geographical (**a**) and geological (**b**) sketch maps of southern Apennines. In (**c**) the geological cross section of the study area.

Recent studies focused attention on the anthropic role as a direct modifier of the landscape [8,17,18]. Active and deliberate anthropogenic actions include, in primis, agricultural and pastoral activities, the building of defensive, agricultural, residential, and industrial structures, canals, roads and railways, mining (subsurface and opencast), ore processing, and waste generation. As a consequence, related to agricultural modification of the landscape, slope erosional processes, hydrogeological instability, and soil loss could happen. These kinds of effect could be considered unintentional [17] consequence of anthropogenic processes, at least until the knowledge of landscape evolution and human impact on the natural system was little known and/or misunderstood. In the present day, we cannot talk about "unintentional consequence".

Human history and human activities on the environment are closely linked to the landscape timing evolution, as demonstrated by the presence of archaeosites, which can testify, at the time, essential modifications of environmental contexts. Geoarchaeology, defined by Huckleberry (2000) [19] as "*the application of Earth Science method and theory to understanding the human past*", for a long time was considered as a sub-discipline of both archaeology and geology [20]. In particular, its goal is to understand the human ecosystem through an in-depth study of the relationships between paleoenvironmental and socio-economic systems in terms of development and evolution [21], so it could play an important role in past societies' environmental context definition. The study of archaeological sites has taken great advantage of data carried out from geomorphological analyses. In particular, as regards the change, over time, of the human use of the territory and of the evolutionary models of settlement (e.g., [22–24]). The definition of geological and geomorphological conditions in archaeological sites is fundamental for the reconstruction of human-environment interactions. Geoarchaeological systems are considered dynamic time-space units [25], inside which archaeosites can be included and analyzed as homogeneous morphodynamic areas. Therefore, in the multicultural context represented by an AMI, the typical topics of disciplines such as geodiversity, geomorphology, biodiversity, archaeology, and geoarchaeology can represent an opportunity to deepen multidisciplinary knowledge when they are conveniently correlated. Geo-heritage management cannot ignore knowledge, management, and protection also of the non-geo contexts in which geo-heritage is an integral part. Geo-heritage management also includes technical and administrative aspects, as well as scientific ones. In this paper, only scientific aspects will be dealt with, referring to the technical-managerial ones to further specific studies.

In the Basilicata Region of Southern Italy, a portion of the Sinni River's catchment area, included in the Sant'Arcangelo Basin (Figure 2), has been chosen with the objective to test and verify the applicability of a multidisciplinary approach and with the aim to improve the territorial promotion and the territorial knowledge and awareness. The analyzed areas include the Sarmento River, the Sinni River's right tributary, and the San Giorgio Lucano and Cersosimo towns. Here, the coexistence of geological, geomorphological, and archaeological contexts makes it possible to deal with different themes. In this area, basic observations on the Geology (sedimentary, stratigraphic structural themes, and so on), Physical Geography (landscape elements, landforms, and dynamics; territorial use, etc.), and specialized analyses on Regional Geology, River Dynamics, Landscape Evolution, Hydrographic Network Evolution of both the Sinni and Sarmento rivers, have been carried out. The analyzed areas can be considered, as a whole, a compound geoarchaeological system (*sensu* [25]) whose evolution over time has been linked to morphogenetic agents and processes, biological and ecological factors, and to anthropic-cultural factors.

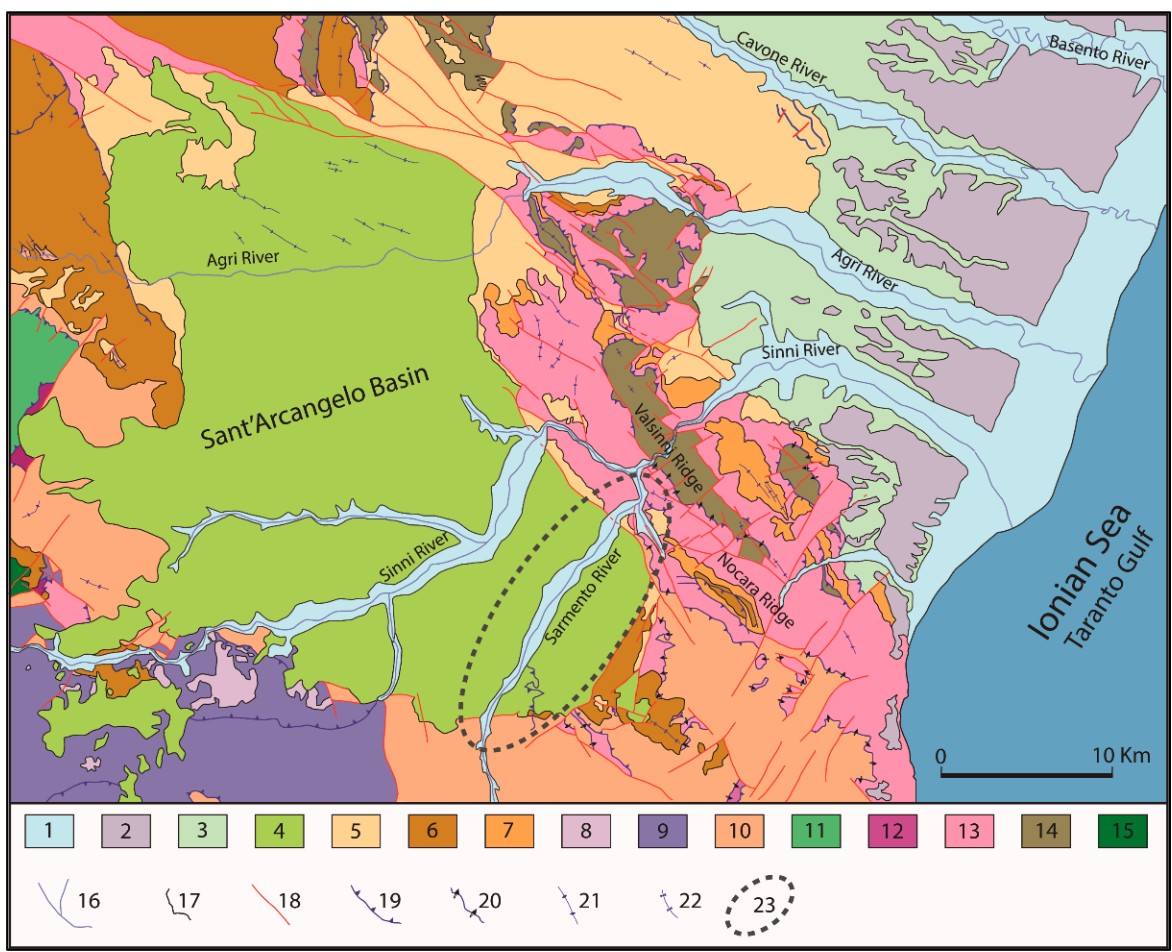

**Figure 2.** Geological sketch map of the study area. Legend: (1) Holocene alluvial deposits; (2) Middle to Late Pleistocene marine terraces; (3) Early Pleistocene subapennine clays; (4) Early to Middle Pleistocene Sant'Arcangelo basin deposits; (5) Early to Late Pliocene Caliandro units; (6) Late Miocene siliciclastic units; (7) Early to Late Miocene siliciclastic units; (8) Timpa Rotalupo tectonic unit (*Palaeozoic*); (9) Frido tectonic unit (*Jurassic-Oligocene*); (10) Nord Calabrese tectonic unit (*Early Cretaceous-Early Miocene*); (11) Monte Raparo tectonic unit (*Late Cretaceous*); (12) Lagonegro tectonic unit: Monte Arioso succession (*Trias-Late Cretaceous*); (13) Lagonegro tectonic unit: Groppa d'Anzi succession (*Trias-Late Cretaceous*); (14) Lagonegro tectonic unit: San Chirico succession (*Trias-Late Cretaceous*); (15) Monte Alpi tectonic unit (*Jurassic-Early Cretaceous*); (16) drainage network; (17) stratigraphic boundary; (18) high-angle fault and undifferentiated tectonic boundary; (19) normal thust fault; (20) reverse thrust fault; (21) anticline line; (22) syncline line; (23) study area.

## 2. Methods and Aims

Italy is one of the world countries with the highest density of historical-cultural-naturalistic heritage (archaeological areas, monuments, museums, parks, and natural areas). Often, a single site can contain different cultural contexts and, too often, individually treated and contextualized in specific cultural contexts and not inserted in wider cultural contexts that would greatly enhance their importance. The division of a territory into Areas of Multicultural Interest (AMI) is the first step to discriminating against its potential and starting a multidisciplinary study in order to create an occasion for territorial promotion and scientific-cultural divulgation. This can be conducted according to a main point of view that can be geological, archaeological, or biological. It does not matter. Among geologists, the geo-point of view is the main one, and therefore they talk about geosites and geomorphosites. We identify an AMI according to a physical geographical criterion, identifying, in this case, the Sinni River's catchment area as such, in which several sites

of multicultural interest (Compound geo(morpho)site *sensu* [4]) are present. We use the term Compound geo(morpho)site with the aim of highlighting how the physical geographical aspects are decisive for the territorial area characterization. To define a compound geo(morpho)site and to discriminate its cultural heritage (geo and non geo, Figure A1) means to have a scientific approach s.s. to its heritage. To define possible correlations between different cultural heritages, making them desirable and reachable for as many end users as possible, means having an extended and resilient scientific approach, taking into account scientific and also additional values (didactic, aesthetic, cultural, and economic). The latter can be developed according to a specific target (Primary Target) and according to the different increasing levels of content, from basic to advanced. Following the first approach, geological, geomorphological, and geoarchaeological contexts are defined and presented for the study area. According to the second approach, educational routes and texts are proposed with different educational levels, starting from basic concepts of geology and physical geography. The main aim of this work is to provide territorial enhancement material for use by a wide audience of end users at different levels of detail, focusing on the Primary Target represented by the Human Role as Geomorphic Agent and The Man-Land use Interplay. A secondary aim is represented by suggesting topics for scientific debate on the role of compound geo(morpho)sites as a geo-cultural dispenser and on the possible best way to do that. In this study case, the coexistence of compound geo(morpho)sites and compound geoarchaeological system represents a further raising for debate on the better way to promote scientific knowledge and citizen territorial awareness. With the aim of encouraging territorial promotion, compound geo(morpho) sites will be indicated with the names of the main towns.

With the purpose of defining a hierarchical terminological order, the terms Main Topic, Topic, Theme, Subject, and Argument, although synonyms are used with the following meaning, starting from global to a more and more specific one. Main Topics represent global concepts (e.g., Evolution, Interplay, and so on) and are the higher hierarchical rank. Topics (e.g., Human Evolution, Landscape Evolution, Environmental Evolution, and so on; Human-Land/Environment/Climate Interplay, Biosphere-Land/Environment/Climate Interplay, Hydrosphere-Land/Environment/Climate Interplay, and so on) represent the next lower hierarchical rank. Within each Topic, different Cultural Contexts (Geo and No geo) can be considered. Within each Cultural Context, it is possible to organize the related contents in three distinct hierarchical levels. These, from the highest to the lowest rank, are represented by Themes (general topics), in turn, composed of Subjects (specific topics) in turn, and characterized by Arguments (particular topics). Figure A2 shows the hierarchical schematization for the Main Topic Interplay, considering the Human Time Scale as Time Target. The Time Target is necessary to contextualize Arguments since geological time has a magnitude, at times, not comparable with biological times. Arguments represent the last considered hierarchical rank until now. The proposed hierarchical organization has the purpose of testing the use of schemes, keywords, and themes for the preparation of a dedicated relational database, also. A database that can be expanded and updated over time, where the material is available for the purposes of territorial enhancement, geoscience divulgation, and territorial citizen awareness.

Extending the recent definition of a Museum (*A Museum is a not-for-profit, permanent institution in the service of society that researches, collects, conserves, interprets, and exhibits tangible and intangible heritage. Open to the public, accessible and inclusive, museums foster diversity and sustainability. They operate and communicate ethically, professionally, and with the participation of communities, offering varied experiences for education, enjoyment, reflection, and knowledge sharing ICOM2022*; available at https://icom.museum/en/news/icom-approves-a-new-museum-definition/ (accessed on 20 March 2022)), whereas a geosite, and in general, the geological heritage, can be considered as an open-air museum. Terms such as accessibility, inclusion, and sustainability can be considered keywords in geo-heritage management. Accessibility, in physical and cognitive terms, as it is not just a matter of ensuring safe use of geo-heritage appropriate to own physical condition but also ensuring adequate cognitive

accessibility according to the cultural level of the end user. Inclusion, both local in society (community participation in knowledge, dissemination, and protection activities), then expanded and widespread, creating the conditions for the geological heritage inclusion in contexts/paths/activities to contrast intolerance caused by judgments, prejudices, racism, and stereotypes. Sustainability in terms of low or zero environmental impact technologies use, proper management of the territory and its bio- (plants and animals) and geo- (soil, landscape, and related elements) components, aimed at protection, conservation, and increasing positive effects on the environment. We are aware of presenting a reductive and incomplete vision of the concepts of accessibility, inclusion, and sustainability which are applicable to a multitude of aspects. For the purposes of this work, we consider the aspects considered sufficient, even if not exhaustive. The studied area will be examined according to the three keywords indicated above after discriminating against its cultural heritage and identifying some possible didactic/tourist paths. Figure 3 shows the procedure followed in the study. Through successive steps, arranged into three stages and characterized by Actions (What) on Objects (Who) based on Criteria/Methods (How), final materials (i.e., Tables, Descriptive Tables, Correlation schemes, etc.), prepared according to the end user needs, have been produced.

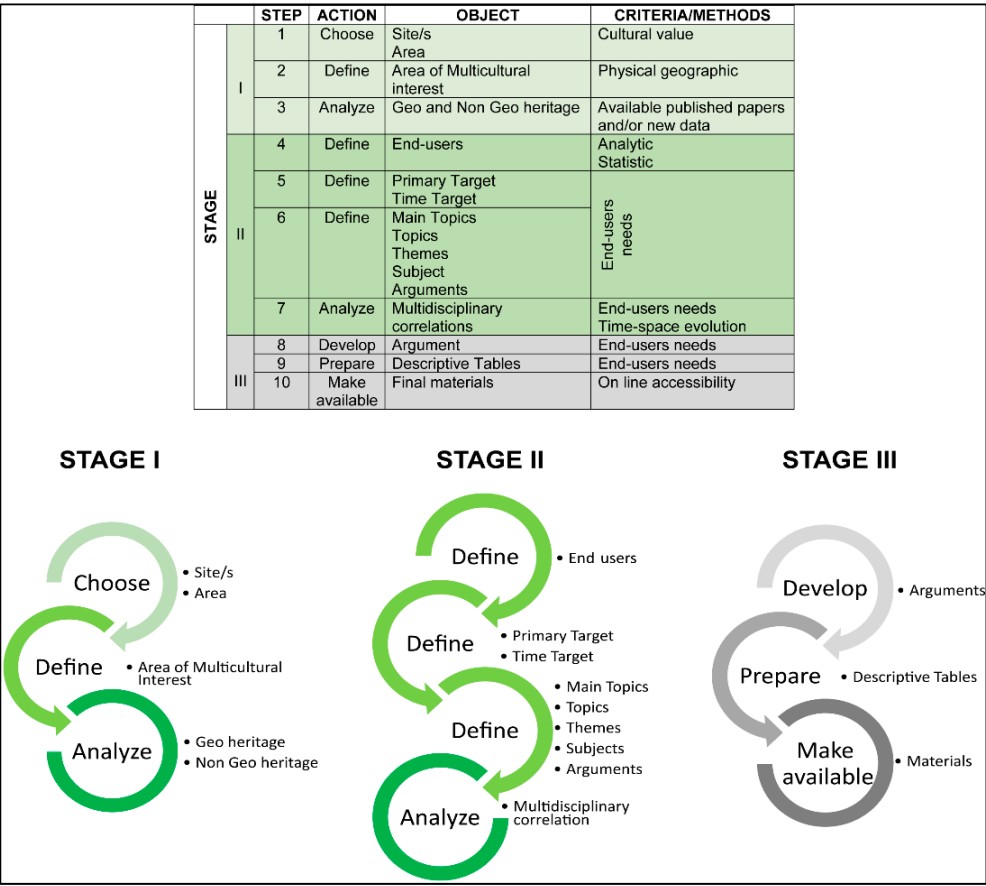

**Figure 3.** Scheme of the adopted methodology in the paper. See text for details.

## 3. General Backgrounds

### 3.1. Geology

The southern Apennines chain is an NW-SE-trending fold-and-thrust belt, formed in a time ranging from late Oligocene to Pleistocene and involved a complex palaeogeographic context characterized by shallow-water carbonate platforms alternating to deep-sea basins. The progressive propagation of the contractional deformation toward the foreland is associated with the development and evolution of a series of younger eastward foredeep basins and by the occurrence of several piggyback/thrust top basins developed on top

of the advancing allochthonous units [26]. In the thrust belt, Meso-Cenozoic deposits are present and mainly represented by shallow-water carbonate units, deep-sea water pelagic carbonate and siliciclastic units; thrust top siliciclastic units; foredeep siliciclastic units. The presence, and structural organization, of several Triassic or Cretaceous to lower Miocene tectonic units and related middle to late Miocene foredeep and/or thrust top deposits and/or related Pliocene to Pleistocene thrust top deposits allowed to discriminate of three distinct zones in the belt, parallel oriented to its NW–SE-elongation axis: the inner, the axial, and the outer. The inner zone corresponds to the Apennines Tyrrhenian side and is represented by Cretaceous to lower Miocene deep-sea pelagic deposits and related lower to late Miocene thrust top deposits (Internal Units *Auctt*.), overthrust on Triassic to early Miocene shallow-water carbonate, and related middle to late Miocene foredeep and thrust top deposits (Apennine platform *Auctt*., including deposits related to carbonate platform s.s. and deposits related to basin transition). The axial zone is characterized by the Apennine platform deposits overlapped on the coeval deep-sea water pelagic carbonate and siliciclastic units (Lagonegro or Lagonegrese-Molisano Units *Auctt*.). The outer zone is characterized by the overlap of chain tectonic units, and related Plio-Pleistocene deposits, on the Plio-Pleistocene Bradano Foredeep units.

The Pliocene to Pleistocene southern Apennines structural evolution was recorded by two main geological elements (Apennine chain and Apulian foreland) and accompanied by the formation and infill of basins located on the former (satellite or piggyback or thrust-top basins), the remains of which are observable in correspondence of the Ofanto, Potenza, Anzi-Calvello, Sant'Arcangelo basins [26–29]. Among these, the Sant'Arcangelo basin was considered a piggyback basin during the Lower to Middle Pliocene and then continued with that of the external foredeep basin (Bradanic trough) during the Upper Pliocene (note: Early Pleistocene after Gibbard et al., 2010 [30]) to Pleistocene. [27]). In the area between Potenza and Sant'Arcangelo basins, an important magnetic anomaly has been recognized. It represents a NE-SW oriented, positive structure and can be interpreted either as a magnetic differentiation or as a portion of the basement uplifted by faults [31,32]. Furthermore, Pleistocene volcanic products outcrop in the northern sector of the Basilicata Region [5,33]. Pliocene to Quaternary strike-slip faults, mainly oriented N120° ± 10° and N50°–60°, are the main tectonic structures affecting the southern Apennines chain [34–36]. That tectonic trend fault were responsible for the genesis of many Quaternary intermontane basins of southern Apennines [37,38], as the faults parallel to the "Pollino Line" [39], or the high Val d'Agri border fault system [40], or the "Scorciabuoi Fault", which borders the Sant'Arcangelo basin to the east [34].

The analyzed area, including the Sinni river (Basilicata region), is located in the frontal sector of the southern Apennines chain and is included in the north-western side of the Sant'Arcangelo basin (Figure 2). The Sant'Arcangelo basin area, one of the larger Plio-Pleistocene basins, is currently delimited to the east by the anticline of the Valsinni–Nocara Ridge (also known as "Tursi-Rotondella Ridge"), to the north by the Scorciabuoi Fault (N120° oriented, affecting both Plio-Pleistocene deposits and the bedrock), to the south-east by an N50° oriented structural alignment, to the west by as the Costa Molina-M. Alpi structural high. To the south and west, Plio-Pleistocene deposits lie on pre-Pliocene units, and locally the overlap is present. The area is characterized by a negative gravimetric anomaly [31,32]. The eastern and western morpho-structural highs of the Valsinni–Nocara and Costa Molina-M. Alpi ridges, respectively, are located in correspondence with structural highs, representing regional scale structural culminations of the buried Apulian Units.

The Sant'Arcangelo basin was filled by a high volume of Pliocene to Pleistocene clastic deposits, ranging from alluvial conglomerates, on the west side, to marine shelfal mudstones, on the east side. Both the stratigraphic succession outcropping in the Sant'Arcangelo basin and its origin and evolution have been analyzed by various authors [27–29,41–51]. The basin has been interpreted as a piggyback/thrust-top basin [26,28,29,43,44,48–57], separated from the "Bradanic Trough" from the structural high of the Valsinni Ridge starting from the Lower-Middle Pleistocene, or as a pull-apart basin connected to roughly NW-

SE-trending left strike-slip fault systems [44,54]. The role of tectonics in controlling the basin fill is generally agreed [26,28,44,52,55]. As regards the chronological attribution of the Pliocene-Pleistocene deposits, it should be noted that in the Geological maps 523Rotondella [50] and 506Senise [51], the Pliocene tripartition (Lower, Middle, and Upper Pliocene) has been used. According to the Pliocene-Pleistocene bound shifted to 2.58 Ma, in the following descriptions, the ages have been calibrated according to [30]. In the study area (Figure 4), according to [48,49], the Pliocene and Pleistocene deposits are represented by the following sedimentary units:

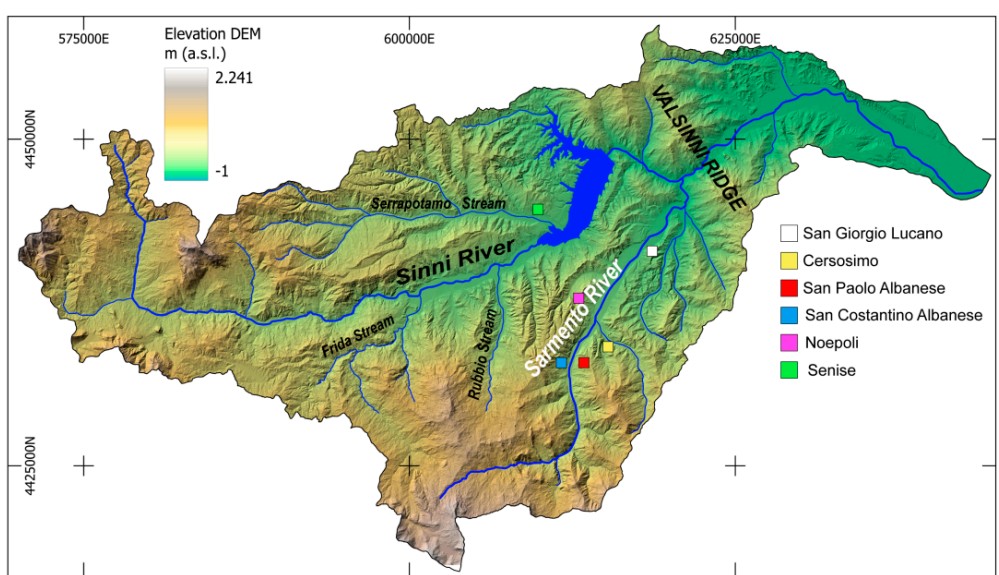

**Figure 4.** Hillshade of the Sinni River catchment basin and the location of the main villages. See text for details.

**Argille subappennine** *Auctt.* **Fm**. (Lower to Middle Pleistocene): hemipelagic deposits consisting of silty marly clays, with sand and gravel levels, related to the Bradanic Foredeep.

**Sant'Arcangelo Fm**. (SAGF, Lower Pleistocene p.p. to Middle Pleistocene): continental alluvial deposits (fan and plain), lacustrine deposits, transitional and marine deposits (delta). The SAGF deposits may be differentiated into the following Synthem units:

*(1) Toppo del Taglio Synthem*, represented by alluvial and slope debris deposits, composed of heterometric carbonate breccias, polygenic conglomerates in the reddish matrix, ochre sands, and red paleosoils; the age is Middle Pleistocene p.p.

*(2) Chiaromonte Synthem* formed of clast-supported conglomerates (alluvial fan deposits) with intercalations of sandy-silty levels, with parallel and crossed stratification (alluvial plain deposits); the age is Middle Pleistocene p.p.

*(3) Francavilla Synthem* composed by clast supported and matrix-supported conglomerates with sandy levels interbedded (alluvial fan and plain deposits), sand, clay, and silt thinly laminated, with plant remains, roots bioturbations, remains of freshwater gastropods, peat and conglomerate levels (lacustrine deposits); the age is Middle-Early Pleistocene p.p.

*(4) Senise/Noepli Synthem* formed by clastic and matrix-supported conglomerates with sandy intercalations, sandstones with conglomerate lenses and clay intercalations (alluvial fan and plain deposits), sandstones with clay intercalations, silty clays, clay and clayey silt (delta deposits); the age is Early Pleistocene p.p.

**Caliandro Group Fm.** (CGF, Early Pliocene–Early Pleistocene p.p.): conglomerates and calcarenites, sands; lagoon clays, clays with levels of sand and calcarenites; diatomitic clays; marly and silty grey-blue clays.

The pre-Pliocene bedrock is represented by the following units:

**Sinorogenic siliciclastic units** (Middle to Upper Miocene): sandstones, clastic, and matrix-supported conglomerates with arenaceous intercalations, marl, and clays.

**Liguride units** formed by: *(1) Frido Fm.* (Cretaceous to Upper Oligocene): highly deformed metamorphic succession consisting of oceanic crust (metagabbros, metadolerites, and metapillows) and continental lithosphere (serpentinized peridotites, metagranitoids, gneisses, amphibolites, granofels, and metacarbonates) covered by deep basin metasedimentary succession (metaradiolarites, calcschists, phyllites, quartzites, and metapelites) and by upper Oligocene calcschists; *(2) Nord-calabrese Fm.* Composed of carbonate and arenite deposits.

**Albidona Fm**. (Upper Oligocene to Early Miocene): arenaceous-pelitic and marly-calcareous turbidites, clays, silt clays, calcareous marl megalayers.

**Saraceno Fm.** (Oligocene p.p.): siliciclastic-carbonate and siliciclastic turbidites.

**Crete Nere Fm.** (Eocene to Oligocene p.p.): shales, silicoclastic and calciclastic deposits.

**Argille Variegate Fm**. (Cretaceous to Miocene): shales, silt clays, silicoclastic and calciclastic deposits.

**Lagonego II units**, here represented by the Numidian Fm. (Upper Oligocene–Early Miocene): siliciclastic and calciclastic deposits, shales, and silt clays.

*3.2. Geomorphology*

The southern Apennine chain is characterized by a line of peaks shifting to south-west, not corresponding, in part, to the regional water division, at least up to the Sirino mountain from where the two lines coincide, continuing towards the south. In the campanian-lucanian sector, the chain's eastern flank has a longer length and a lower average gradient than the western flank [58]. From a geomorphological point of view, the southern Apennine chain can be divided into three zones: inner, axial, and outer. The Sant'Arcangelo basin is located in the outer zone, west of the regional watershed. Two main rivers, the Agri and the Sinni rivers flowing eastward to the Ionian Sea, are present. The Sinni River (Figure 5) is one of the major rivers in the Basilicata region, flowing out of the Sirino Massif (2.005 m of elevation a.s.l.) to the eastern side. It runs almost 100 km before reaching the Ionian sea, crossing the Metaponto coastal plain. The catchment basin of the Sinni River has an area of 1360 km$^2$, of which about 16% reaches altitudes between 900 and 1200 m and over 54% has an altitude higher than 600 m, while 30% is lower than 300 m. The eastern and south-eastern portions of the Sinni catchment include the western and north-western slopes of the predominantly carbonate mountains (Sirino Mts., Alpi Mts., Lauria Mts. Pollino Mts.); the central sector is characterized by hills mainly consisting of Plio-pleistocene silico-clastic deposits. Finally, the eastern sector includes, from W to E: (i) a mountainous to high-hilly landscape comprising the Valsinni ridge and consisting of Ceno-Mesozoic quartzarenite, limestone, and clayey deposits; (ii) a low-hilly landscape, where the Plio-Pleistocene clastic deposits outcrop; (iii) a morphologically flat area, close to the coastal line, characterized by the presence of alluvial successions and coastal plain successions. By starting from the springs going towards the estuary, the Sinni river is fed, on its hydrographic right, mainly by (i) sources related to the Monte Sirino hydrostructure, and, secondarily, to the Lauria Mountains hydrostructure, (ii) the Frida Stream and its tributary such as the Peschiera Stream, fed by the Pollino hydrostructure and by the M.Caramola hydrostructure, (iii) the Rubbio Stream, (iv) the Sarmento River; on the hydrographic left the main tributaries are (v) the Cogliandrino creek, (vi) the springs related to the Monte Alpi hydrostructure (La Calda and Caldanella springs),(vii) the Serrapotamo Stream and (viii) the Fiumarella di Sant'Arcangelo Stream.

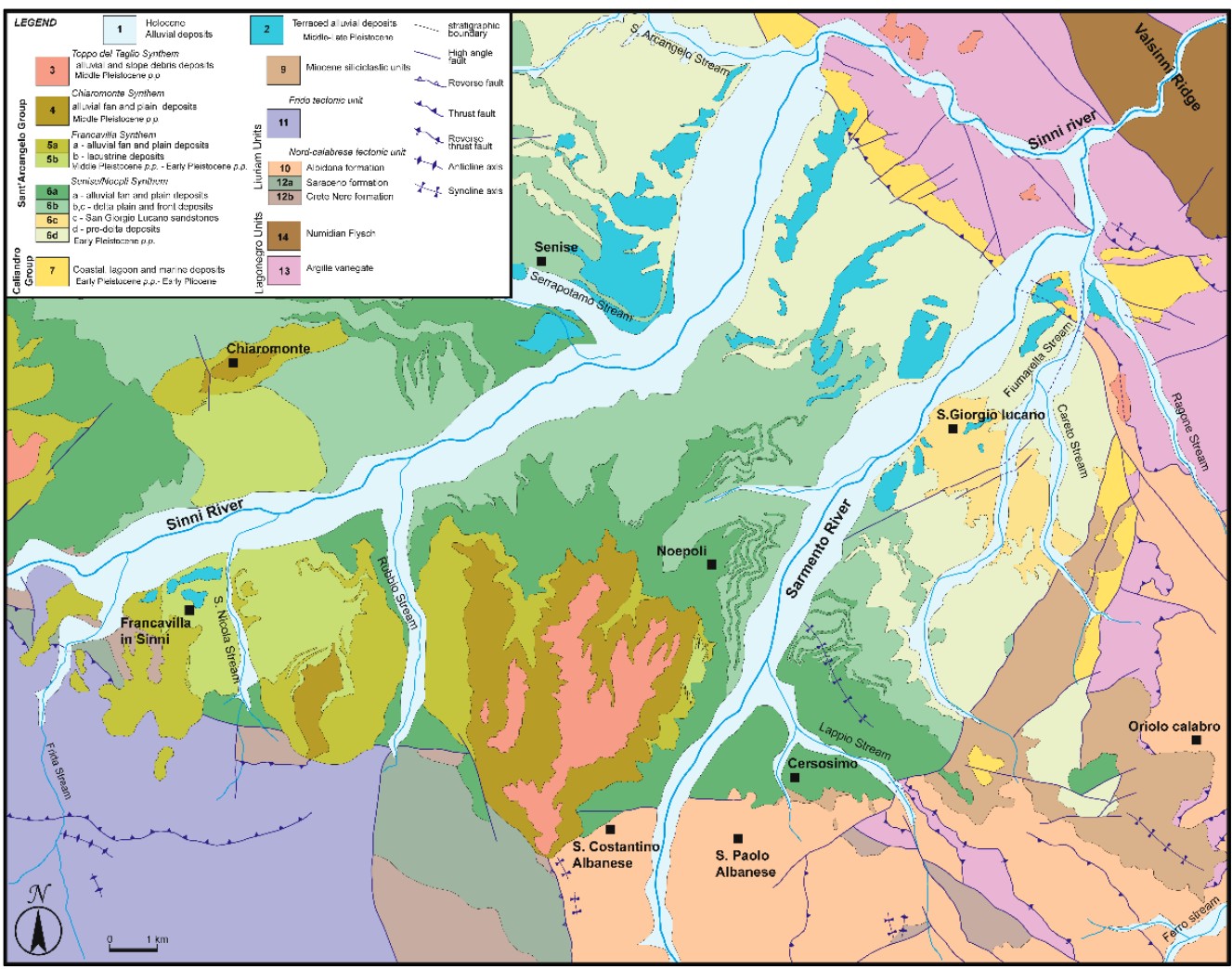

**Figure 5.** Geological sketch map of the San Giorgio Lucano area and surroundings.

The Sinni River crosscuts the NW–SE oriented southern Apennines fold-and-thrust belts, producing a tectonically controlled drainage pattern [59]. The river shows different trends of direction, following the main morphotectonic alignments. Firstly, it follows a NE-SW direction along the Sirino Massif eastern side, then it follows a WNW-ESE direction and continues along an SW-NE direction until reaching the Cogliandrino lake, which it follows an NW-SE direction. Afterward, until Francavilla sul Sinni town, the Sinni River shows an undulating trend, following NW-SE ad NE-SW directions. From the Francavilla sul Sinni town to the Monte Cotugno lake, the river shows mainly ENE-WSW and NE-SW directions. After the Monte Cotugno dam, the Sinni watercourse begins to incide vertically the Valsinni ridge taking an NW-SEW up to the confluence with the Sarmento River, after which the Sinni river flows along a WSW-ENE direction. The watercourse's last part, up to the estuary, follows an NW-SE direction. The Sinni river's longitudinal profile is characterized by knickpoints, gorges, and floodplains [59] (Figure 6). Two different landforms stand out in the present-day Sinni fluvial valleys: (i) deep gorges carved both in bedrock than in clastic deposits and (ii) large floodplains filled by fluvial and fan deposits. The floodplains show braided or meandering channel features with a flat longitudinal profile, and the valley flanks are characterized by the presence of several fluvial terraces [59].

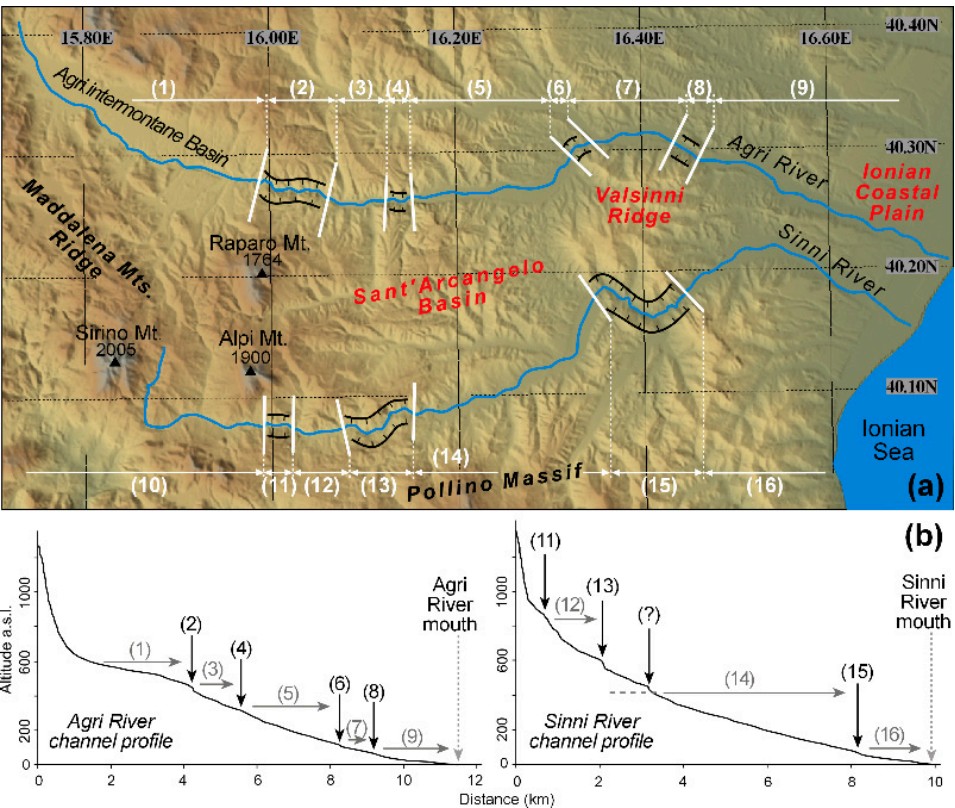

**Figure 6.** Gorge valley and floodplain landforms (**a**), and longitudinal channel profiles (**b**), recognized in the Agri and Sinni river valleys (modified after [59]).

### 3.3. Geoarchaeology

Until the Mesolithic time, human activity was mainly linked to hunting and gathering and had a relatively minor impact on the environment; starting from the Neolithic, with the birth of agriculture and livestock, man began to interact in an increasingly incisive way with his environment. Several paleoanthropological data indicate a significant population increase in the transition from harvesters to farmers [60], an increase that, over time, has exerted ever greater pressure on the territory through deforestation to obtain fields to cultivate and pastures. In the human evolutionary path, water plays a fundamental role, both in the aspects related to the presence of springs (ancient settlements are characterized by springs within or in the immediate vicinity), lakes, and navigable waterways (communication routes and exchange between coastal areas and the interior), both as a primary asset necessary for species survival.

In the present-day Region of Basilicata (Lucania), the first traces of human presence date back to the Lower Paleolithic, when rivers and lake basins represented environmental niches particularly favorable to the spread of rich and varied flora and fauna and, therefore, to the human presence (basins of Atella, Melfi, Venosa, and Mercure, for instance). The study of the territory over time [61] (see, among others, Atti Convegni sulla Magna Grecia from 1962 to 2015, http://www.istitutomagnagrecia.it/pubblicazioni/atti-dei-convegni/ (accessed on 20 March 2022); SIRIS, https://edipuglia.it/siris-open-access/ (accessed on 20 March 2022); Consiglio regionale della Basilicata, Quaderni http://www.old.consiglio.basilicata.it/pubblicazioni/main.asp (accessed on 20 March 2022)) has shown traces of human presence and attendance, represented by farms, furnaces, cult and burial sites, fortified sites and ancient connecting roads, concentrated, as well as along the coastal strip, mainly near springs and streams, on heights and reliefs, to control the territory and these itineraries. Important waterways, such as Bradano, Basento, Cavone, Agri, and Sinni rivers, navigable from ancient times to the Middle Ages, represented important communication

routes between the coastal Ionian Sea areas and the hinterland, and, through foothills and passes, also with the Tyrrhenian coast [62,63]. Along these waterways, the presence of burial sites highlights their attendance already during the early Iron Age (IX-VIII secolo B.C.) [62]. As noted by Quilici and Quilici Gigli (2003) [64], along the Sinni River and its tributaries, the Sarmento and the Serrapotamo rivers, the territory is:

(a) characterized by (i) large flat areas and plateaus, (ii) water availability (in ancient times), (iii) the presence of a waterway whit navigable long stretches;

(b) protected by (iv) a narrow gorge (Valsinni Ridge) and with (v) heights that allowed close control of the territory;

(c) connected with(vi) two important commercial cities located at the mouth of the Sinni River, Siris and Heraclea villages;

(d) located on (vii) the communication route between the Ionian Sea and the Tyrrhenian Sea [63].

The discovery of several necropolis and farms related to the Greek-Lucanian period [65] testifies to the human attendance and the agricultural use of the territory. The agricultural cultivation had to be concentrated not only in the flat areas of the Ionian coast, occupied by the Greek colonists but also in the heights and on the more internal plains, presumably occupied by indigenous peoples. Together with agricultural cultivation, breeding was presumably one of the activities present on the territory. Agricultural cultivation and stockbreeding are commonly associated with increased deforestation to obtain arable land and pasture, deforestation which is related to an increase in the rate of soil erosion. Studies carried out in Greece show how excessive exploitation of the territory by ancient Greek may be the basis of the current environmental degradation [66,67]. Similarly, it can be assumed in southern Italy (*Magna Grecia*) a similar approach to the territory by Greek settlers [68]; although focusing on the ancient past diverts attention from the recent past, when knowledge and technology should have guaranteed, if not the recovery, at least in do not worsen the environmental conditions of many territories. According to some authors [69–72], the intensification of agricultural cultivation has favored the erosion of the soil and the accumulation of sediments on the valley floor. Other authors believe that the anthropic factor is irrelevant compared to the climate [73–75].

## 4. Results and Discussion: Compound Geomorphosites

Within the Sinni River's catchment basin, several suitable sites to be considered as Compound Geomorphosites are present, as suggested by [4] and applied by [5]. Taking into account all potential sites is beyond the goal of this work, so we will only consider the San Giorgio Lucano area, just hinting at other sites nearby. Other suitable sites to be considered as Compound Geomorphosite are the following areas: Cersosimo; Noepli; San Paolo Albanese—San Costantino Albanese; Epicopia; Latronico, Chiaromonte; Monte Alpi; Valsinni Ridge.

### 4.1. San Giorgio Lucano

San Giorgio Lucano is a small town located on an oriented NE-SW engraved ridge on the Sarmento River's hydrographic right before it flows into the Sinni River (Figures 4 and 5). The ridge is bordered towards NW by the Sarmento River and towards SE by the Fiumarella Stream; in correspondence with T.mpa Ciucca (669 m of elevation a.s.l.), the ridge changes direction towards SE, and the Lapio Stream separates it from NW-SE oriented ridges on which the towns of Cersosimo and San Paolo Albanese are present.

- Geo Heritage

San Giorgio Lucano gives its name to a large sandy lens interspersed in the Sant'Arcangelo basin clayey succession (Sands of San Giorgio Lucano *Auctt.*). The sandstones consist of (i) moderate to well-cemented quartz-rich planar or trough cross-bedded sandstones and (ii) silty sandstones with rare intercalations of clays. Sandstone bed thickness ranges from a few centimeters to some decimeters, while clays form centimetric horizons; their body

geometry varies from lens shaped to tabular. Sandstones present: (i) marked erosional basal surfaces; (ii) bed-parallel lamination and cross lamination; (iii) ripples and truncation surfaces; (iv) bioturbation and borrows; (v) macrofossils. The clayey succession consists of blue-grey marly clays with indistinct stratification and pseudo-conchoidal fracture; sandy-silty levels, lenticular levels of sand, and abundant fossil remain are present (Figure 7). In this work, the nominative distinction is maintained even if they are considered it is corresponding to the arenaceous facies belonging to the Senise/Noepli synthem. In the San Giorgio Lucano area, several outcrop rock sites are available, where it is possible to observe the arenaceous, sandy, and clayey lithologies and the presence of fossil remains.

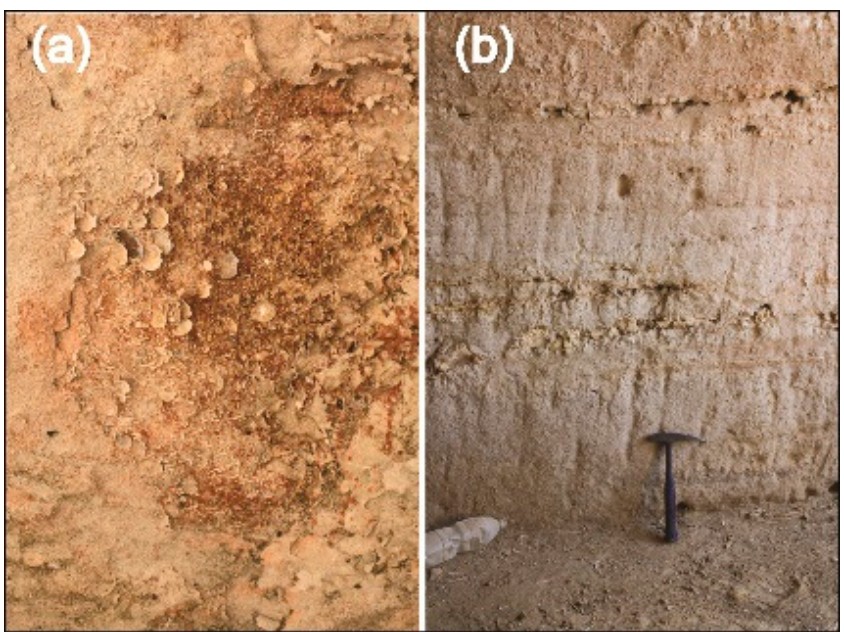

**Figure 7.** (**a**) Detail of fossil remains; (**b**) detail of stratigraphic layers inside the hypogea of the San Giorgio Lucano.

- Non Geo Heritage

Non Geo Heritage is represented by several cultural contexts (Figure A1); among these, for the purpose of this work, it was decided to consider the archaeological context as a priority.

The presence of several archaeological areas (burial sites, furnaces, and farms) related to the Greek-Lucanian period [65] testifies to the human attendance and the agricultural use of this territory. Several archaeological finds are currently observable at the National Archaeological Museum of Potenza, where it is possible to see, among other things, the fragments of armor [76]. North and South of San Giorgio Lucano town, long stretches of paved mule tracks connecting San Giorgio Lucano to Valsinni and Cersosimo villages are still visible. Several hypogea (Figure 8), used in the past either as home or recovery for domestic animals and foodstuffs deposits, are present [77]. According to historical data, hypogea were supposed to be excavated by Basilian monks between the VIII and XI centuries [78,79], who fled from the Byzantine Empire to southern Italy because of Emperor Lion III Isaurico's iconoclastic persecution, which started in 726 AC.

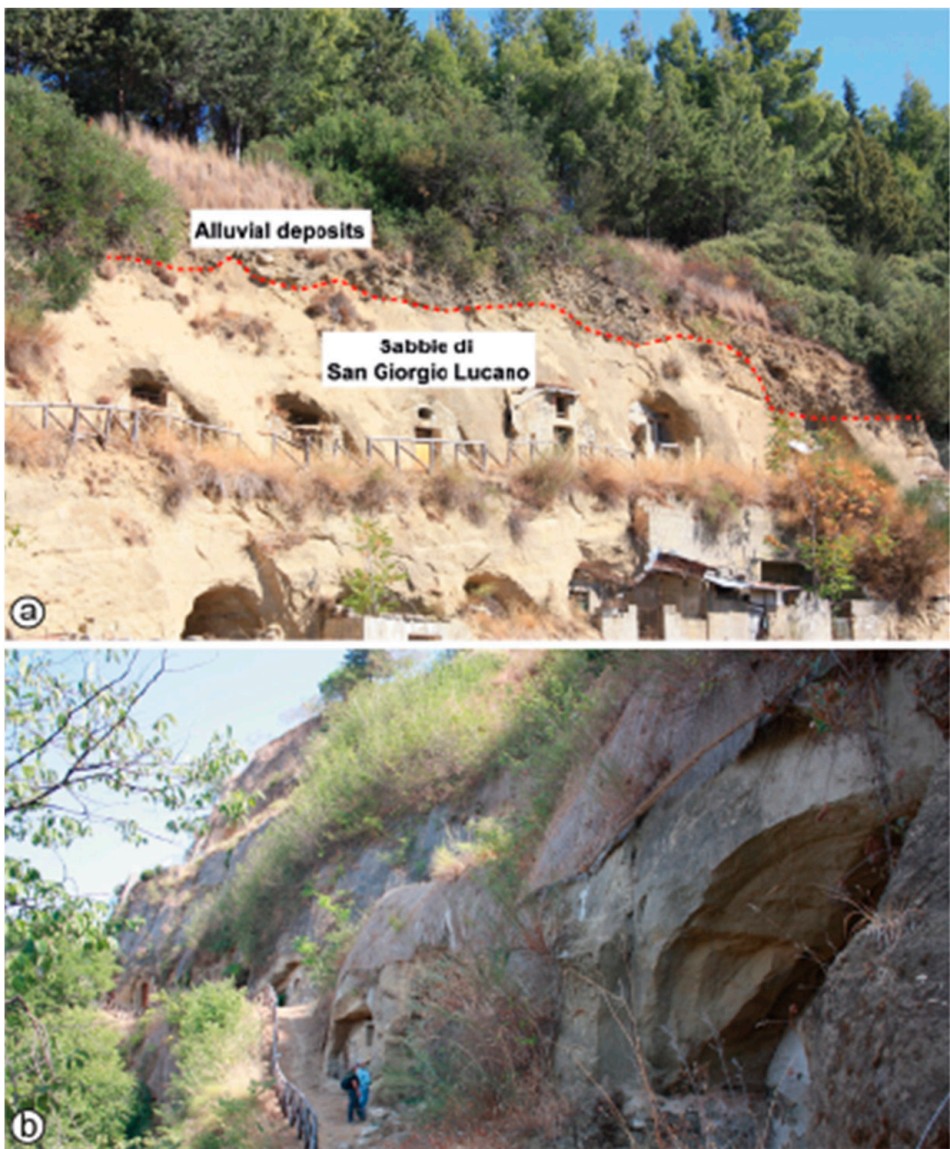

**Figure 8.** Details of the Timpa Selvavecchia (**a**) and Le Timpe (**b**) hypogea (modified after [77]).

In the Basilicata Region, the Basilian monks created small communities, choosing isolated, well-hidden, and protected sites as places to live, preferring the presence of natural cavities to be used as places of worship. In the area of San Giorgio Lucano, there were cavities already used previously as storage or shelter for animals, probably. Hypogea spread over a large part of the municipal area. They are excavated in the Sands of San Giorgio Lucano arenaceous lithologies and are mainly concentrated in the Le Timpe area, a W-E oriented slope facing N toward the river Sarmento, and Timpa Selvavecchia area, an SW-NE oriented sloper facing SE toward a deep SW-NE oriented incision separating the relief of Timpa Selvavecchia from that of San Giorgio Lucano. In the Le Timpe area, the caves penetrate up to 15 m in the sandstone; two or more rooms are present, and their size progressively decreases toward the bottom, with a telescopic shape view in plain (Figure 9a). The caves at Timpa Selvavecchia commonly consist of a single, rectangular-shaped cell. A masonry of stones supporting the entrance door architrave, generally hosting a small window for lighting the room, is present. Along the internal walls, carved niches housing agricultural tools and food are present. Outside the cave, a small yard used for poultry is often present (Figure 9b).

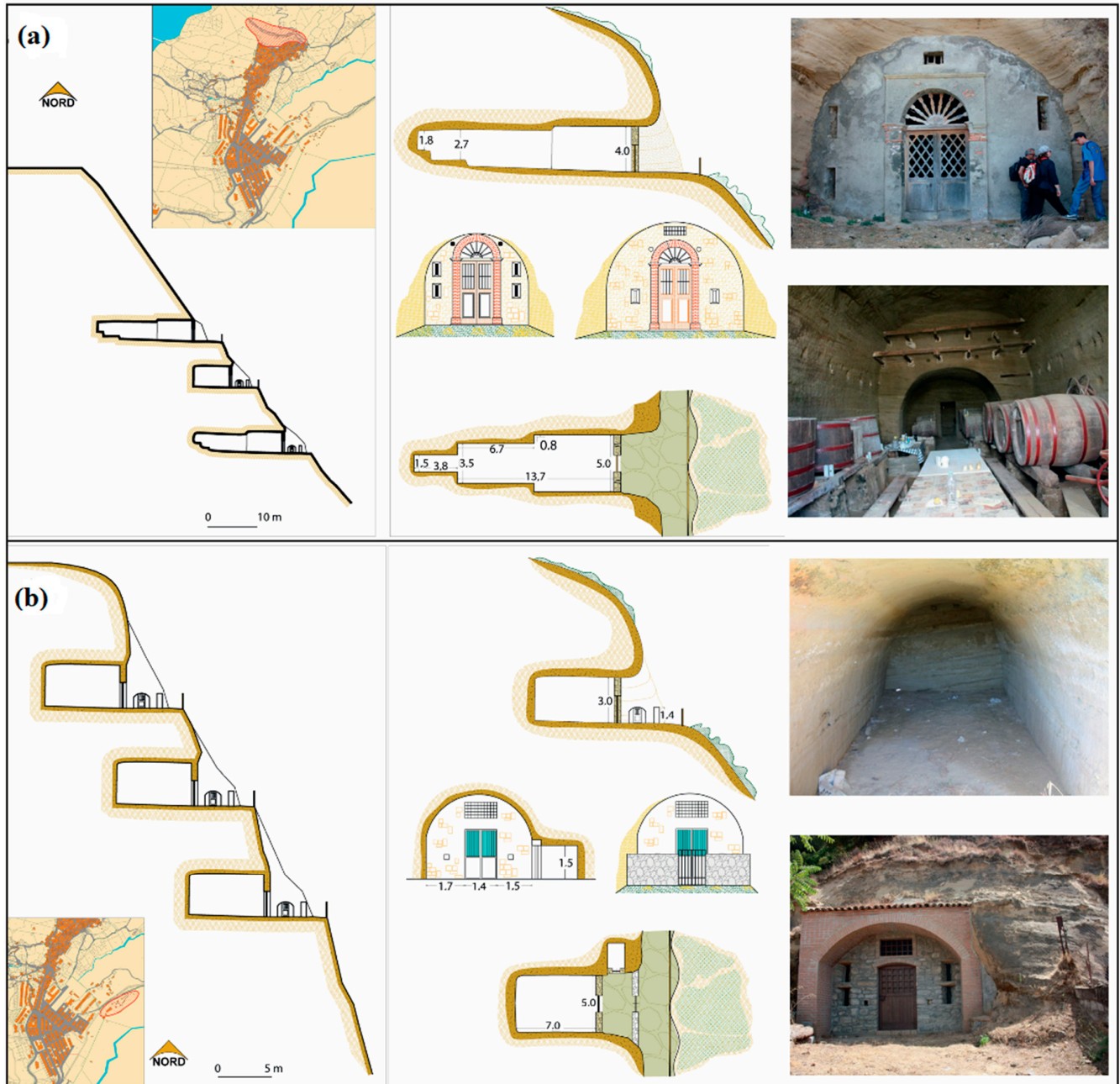

**Figure 9.** Architectonic cross-section of the Le Timpe (**a**) and Timpa Selvavecchia (**b**) hypogea (modified after [77]).

### 4.2. Cultural Contexts Schematization

The main geo-themes and non geo themes here considered are presented in Figure A2. Furthermore, for the purposes of the work, the Main Topic is represented by Interplay, and Topics, selected considering the Human Time Scale as Time Target, are represented by Man-Land use Interplay, Man-Environment Interplay, and Land-Climate Interplay. Cultural contexts, not exhaustive of the area potential, represented by History, Geology, Physical Geography, and Biology, are contextualized within Topics (Figure 10). The apparent theme repetition in Figure A2 actually exhibits different points of view on the theme, namely the Topic point of view, i.e., Man-Land Use/Environment Interplay or Land-Climate Interplay.

Conceptually, it is possible to foresee deepening different levels of topics, starting from the basic ones (Level I) to the advanced ones (Level IV), after which more advanced

levels are possible (level V and more) (Figure 10). In a previous paper, we considered Educational both the scholar end user and the tourist end user [5]. At the current state of research, we believe we must distinguish between Educational, Dissemination, and Tourism. A multi-level schematization can be used both at the Educational/Training level, then at the Dissemination level, or the Tourism level (Figure 11). Arguments must be prepared and presented in a form compatible with the level, obviously. The use of terms and content appropriate to the level is essential for the topic's proper dissemination; using language, and content, in an excessively academic style for a level below III could lead to a dissemination contraction rather than to its enlargement, for example.

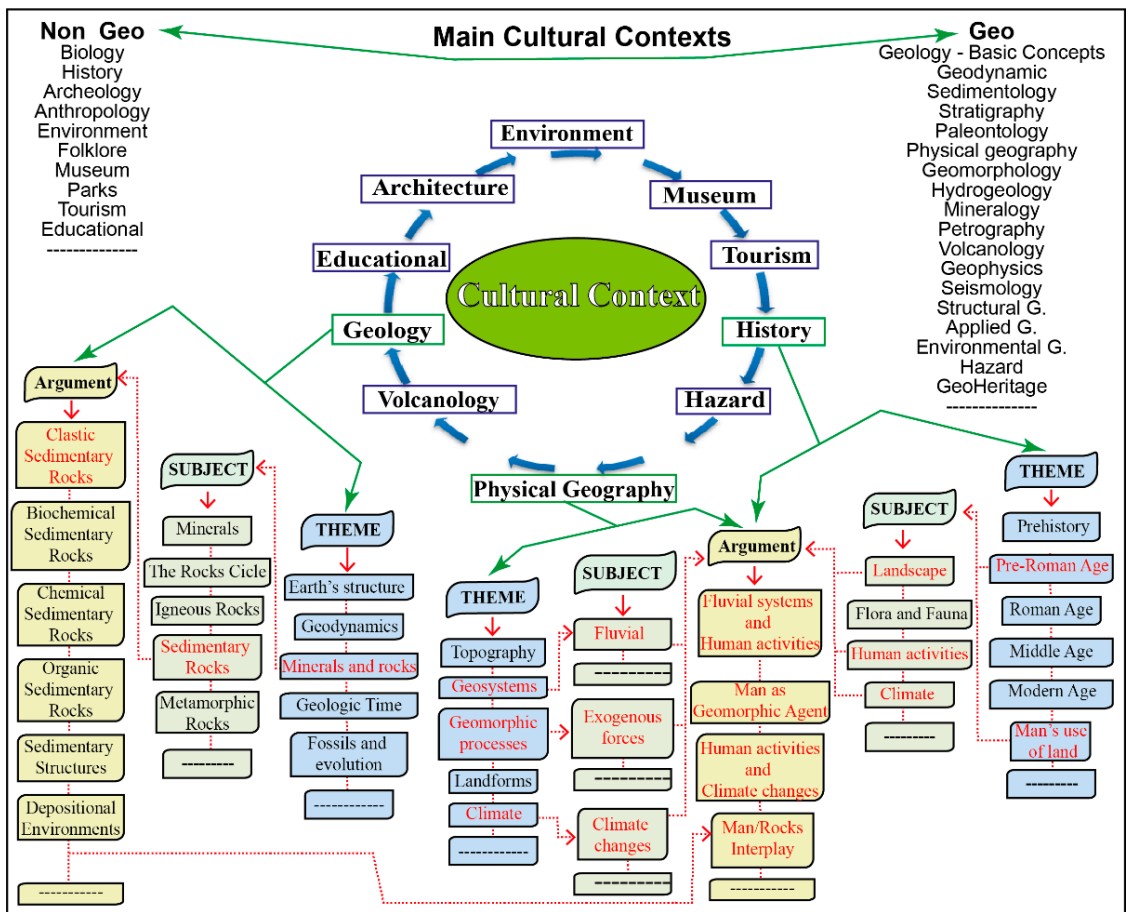

**Figure 10.** Scheme of cultural context showing the hierarchization and correlation of Theme, Subject, and Arguments related to History, Geology, and Physical Geography. The environment in Non Geo context is referred to as its biological matrices. In red Themes, the Subject and Arguments considered for the analyzed area are highlighted.

As regards the educational purposes: (i) the basic level (Level I) is related to the pupils who face for the first time the problems relating to the physical territory, the environment, and the relationship between man and the latter (primary schools); (ii) the medium level (Level II) is related to students who have acquired the basics of scientific knowledge (Secondary School): (iii) the advanced level (Level III) is related to university students. The text contents associated with the various levels, presented as Descriptive Tables, are prepared according to a didactic excursion (Field Trip). The Descriptive Tables are organized into modules: (i) a first module dedicated to preparatory activities to be carried out before the Field Trip, such as the definition of concepts and problems; (ii) a second module dedicated to the Field Trip and related activities; (iii) a third module dedicated to activities to be carried out after the Field Trip. from advanced level IV onwards, end users and objectives change. Not only didactic or informative material but research ideas

designed for post-graduate students and researchers are supposed in order to increase the discussion on the previous considerable amount of works available in the light of current research and to promote new research. Descriptive Tables are supposed to have an Introduction related to the generic Theme and Subject, a Bibliographic Background related to the Argument, and a part of possible themes to be developed.

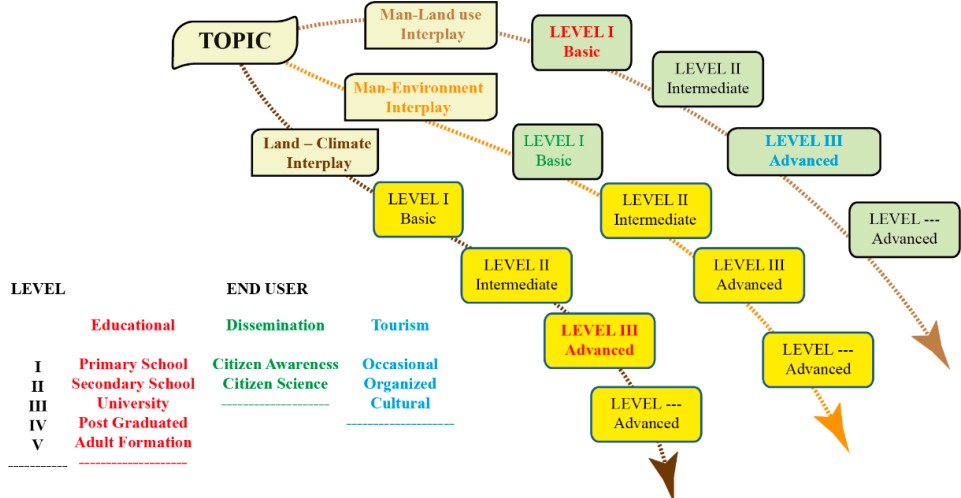

**Figure 11.** Flow diagram showing the level increase in contents for educational, dissemination, and tourism end-user.

Concerning the Dissemination purposes, goals are concerning the increasing Citizen Awareness, participation in Citizen Science, and all the activities, all actions aimed at making scientific and research topics more popular and popularized. Aspects related to Tourism purposes are considered separately from the previous ones while sharing their contents. For simplicity, we schematize tourists as occasional (Level I), organized (Level II), and cultural (Level III) tourists; this topic is currently the subject of in-depth analysis for further paper to discriminate the best way to prepare material for tourism purposes, material that must be adequately proposed while retaining its scientific value, of course. As mentioned above, contents are presented as a Descriptive Table, organized into three modules related to the Level; all these contents should be supplemented by in situ presence of appropriate explanatory billboards. Such as example, the following contents are exposed:

**Educational**

*Level I* Man-Land use Interplay
*Level III* Man-Land use Interplay

Man—Environment Interplay

Land—Climate Interplay

**Tourism**

*Level I* Man- Land use Interplay

Level I Educational contents are presented as a Descriptive Table (Figure A3, Descriptive Figure A1). Following the Cultural Trail, further activities are possible in the classroom, such as a report on the observed elements (landscape, environment, rocks, fossils, cavities) or searches on similar sites in Italy and abroad, and so on. The Topic is Man-Land use Interplay, included in several Cultural Contexts, among which the following are considered (Figure 10):

- Geology

  Theme-Mineral and Rocks

  Subject-Sedimentary Rock
  Argument-Clastic Rocks, Man/Rocks Interplay

- History

  Theme-Pre-Roman Age, Medieval Age, Mans's use of land

  Subject-Landscape, Climate, Human activities

  Argument-Man/Rocks Interplay, Fluvial System and Human activities
  Human activities and Climate changes.

Level III Educational contents are arranged for students following a Physical Geography course (Bachelor of Science in Geology), during which field trips are planned. Contents are presented as a Descriptive Table (Figures 12 and A5, Descriptive Figure A2).

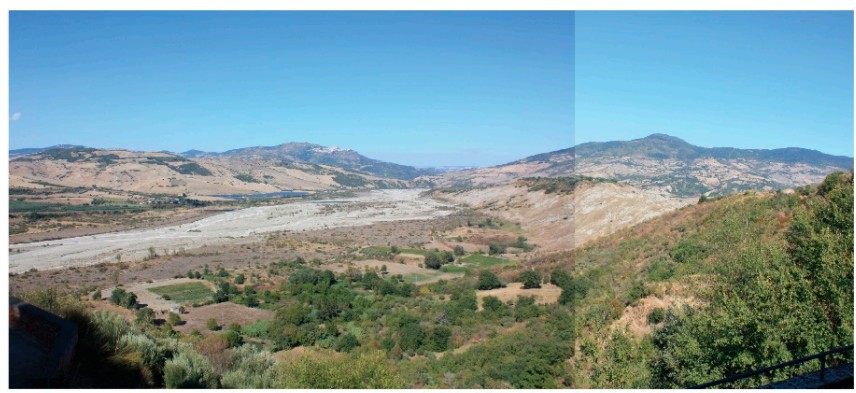

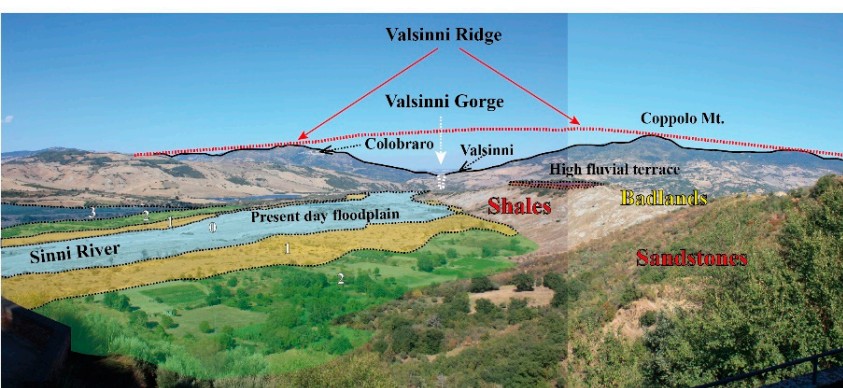

**Figure 12.** Panoramic view from the San Giorgio Lucano hypogea. See text for details.

Topics Man-Land use Interplay, Man—Environment Interplay, and Land—Climate Interplay are related to Physical Geography Context (Figure 10):

- Physical Geography

  Theme–Geosystem

  Subject–Fluvial Geosystem

  Argument–Fluvial Geosystem and Human activities

  Theme–Geomorphic processes

  Subject–Exogenous forces

  Argument–Man as geomorphic agent

  Theme–Climate

  Subject–Climate changes

  Argument–Climate changes and Human activities

Level I Tourism contents are arranged as brochures that occasional tourists can find in public places such as bars, petrol stations, newsstands, and tourist information centers (Figure A6, Descriptive Figure A3), with contents useful to entice occasional tourists to visit the places described.

*4.3. Geoheritage Management*

In an Area of Multicultural Interest, the knowledge of heritage (geo, bio, archaeo, cultural, and so on) and how it can be composted and used for territorial enhancement, science, and geoscience divulgation and territorial citizen awareness represents the first step for its management. The multi-level schematization previously used and described distinguish between Educational, Dissemination, and Tourism Level and refers only to the disclosure aspects intended for the public, for end users falling into those fields. A different speech must be made for the aspects related to geo-heritage technical-administrative management; these aspects will only be blinded in this paper, mainly set on didactic/informative themes.

Geoheritage management should not be considered as a stand-alone object but inserted in a wider context that takes into account the physical environment in which it is inserted (analyzed at an appropriate scale) and the other elements (no-geo heritage) that are part of it and, as part of the same environment, suffer the same influence and can affect it. Proper management of geo heritage is a wide-ranging management that should include the following actions. Define: (i) all the elements that make up the territorial heritage; (ii) the possible interconnections between the various elements; (iii) heritage critical issues in the short-, medium-, and long-term evolution time. Create: (i) content and paths where the geo element is the fulcrum around which the other elements rotate. Evaluate: (i) current status; (ii) possible short, medium, long-term evolution; (iii) best practices for protection and conservation. Planning, in the short, medium, and long term: (i)maintenance, restoration, and consolidation; (ii) updating of studies on the present heritage; (iii) measures to involve local realities; (iv) best practices for dissemination; (v) heritage dissemination using modern technologies, updating following the technological rising.

Regarding Geoheritage accessibility, inclusion, and sustainability, we could talk about a basic level to be guaranteed for each of the key terms, simplifying, simplifying a lot. In the analyzed area, the San Giorgio Lucano hypogea represent an example of limited physical accessibility. The paths are not suitable for users with limited or reduced mobility; cognitive accessibility is one of the goals of the present paper and is currently being worked on. The hypogea complex could be considered an inclusive historical and environmental site. The use (and abuse, unfortunately) of the cavities over time testifies to a close relationship between the environment and human needs. Inclusive, therefore, in the "*physical*" meaning of the term "*integrated in a cultural and environmental context in a given historical period.*" In the modern cultural and social sense of the term inclusive, the hypogea complex is a potential inclusive site if adequately prepared for inclusion through interventions aimed at its conservation, enhancement, and dissemination. Moreover, from a sustainability point of view, potential sustainability is present if all necessary restoration and safety measures are carried out with a green sustainability view. As pointed out by [77], previous maintenance and restoration interventions have led to a momentary and apparent situation of improvement and stabilization. The stabilization measures carried out have not led to the expected results. Indeed, the stability conditions of the entire slope have worsened.

## 5. Concluding Remarks

Geo-heritage (but this applies to heritage in general) is an opportunity for territorial enhancement to promote scientific knowledge and territorial awareness of citizens. It is also an opportunity to introduce and address topical issues such as climate change and the human role in territorial changes and climate change. In geo-heritage management for dissemination to the public (this applies to all types of heritage, again), the scientific community has the task of analyzing the various aspects and scientific content, seeking, evaluating, and proposing possible correlations with other cultural aspects; to use language and content appropriate to the context to which they will be dedicated; to make the content available using modern technologies. A portion of the Sinni River's catchment area (Basilicata region, Southern Italy), including the Sarmento River, one of its right tributaries, has been chosen to test and verify the applicability of a multidisciplinary approach in order

to improve the role of geological themes as a fulcrum around which other cultural elements may rotate, creating new cultural/educational/touristic realities and opportunities.

In the San Giorgio Lucano area, the coexistence of geological, geomorphological, and archaeological contexts makes it possible to deal with different themes and different levels of approach. A multi-level schematization can be used to distinguish between Educational, Dissemination, and Tourism Levels, in which Arguments must be prepared and presented in a form compatible with the level obviously. The use of terms and content appropriate to the level is essential for the topic's proper dissemination; using not appropriate language and content could lead to a dissemination contraction rather than to its enlargement. The analyzed examples show how the same object (with its many geo and not geo-cultural components) can be presented in different ways to different end users, modulating the content according to the needs of these. Studying and analyzing a territory from a scientific point of view *sensu strictu* and also expanded, modulating content and themes according to what will be the final use, helps to increase knowledge and awareness of the many cultural aspects that characterize several areas, more or less large, in Italy as abroad. The main critical issues encountered in writing this paper concern the difficulty of finding the necessary material, the different criteria followed in the processing of data, and the use of keywords. In particular: papers are not always easily available online; the approach to data (in terms of collection, cataloging, processing, and restitution) varies according to the cultural context considered; sometimes key, words are used with different meanings depending on the cultural context taken into account.

The multi-disciplinary and multi-level approach to geo and not geo heritage can be one of the fundamental keys in the management of the heritage. Management that must concern both the geo and the non geo heritage, when these are coexistent and closely connected, as they are parts of the same environment, suffer the same influence and can affect it; and, above all, their modification can positively or negatively affect the whole environment. Often, faced with a substantial and solid geological, naturalistic, landscape, and archaeological background that would allow the inclusion of sites in different educational/cultural/tourist routes, areas of interest are examples of heritage unsuitable management. To study an area by discriminating against all its potentialities is not enough if appropriate actions of protection, valorization, and diffusion are not associated. Even if the San Giorgio Lucano hypogea represents an example of limited accessibility, inclusivity, and sustainability, its potential is very strong. The proximity to archaeological sites and museums (Metaponto, Policoro, Nova Siri), the Ionian beach, the characteristic villages (Colobraro, Valsinni, Cersosimo, etc.), its location along the route that leads inland and towards the Pollino National Park, make San Giorgio Lucano (and its hypogea) a potentially interesting destination for territorial promotion.

We mentioned a database and the use of tools to make information easily accessible. Currently, the work group is developing key terms, themes, and contexts that can be used as main keys in the database. Through the main keys, the end users will be able to access the contents, choosing according to their needs. Contents presented in this work are examples of free material that will be available.

**Author Contributions:** Conceptualization, E.P. and M.B.; methodology, E.P., M.B. and S.I.G.; software, E.P.; validation, E.P., M.B. and S.I.G.; formal analysis, E.P., M.B. and S.I.G.; investigation, E.P., M.B. and S.I.G.; data curation, E.P., M.B. and S.I.G.; writing—original draft preparation, E.P. and S.I.G.; writing—review and editing, E.P., M.B. and S.I.G.; supervision, E.P., M.B. and S.I.G. All authors have read and agreed to the published version of the manuscript.

**Funding:** This research was funded by M. Bentivenga, grant number "R.I.L. 2020" and by S.I. Giano, grant number "R.I.L. 2020".

**Institutional Review Board Statement:** Not applicable.

**Informed Consent Statement:** Not applicable.

**Data Availability Statement:** Not applicable.

**Acknowledgments:** We wish to thank the Editor for the useful comments, A.Y. Karunarathne, and three anonymous referees for their critical suggestions, which improved the paper.

**Conflicts of Interest:** The authors declare no conflict of interest.

## Appendix A

| MAIN | | | | | |
|---|---|---|---|---|---|
| GEO | **CULTURAL CONTEXTS** | **THEMES** | NON GEO | **CULTURAL CONTEXTS** | **THEMES** |
| | Geology | Earth's structure | | Biology | |
| | | Minerals and Rocks | | History | Prehistoric Age |
| | | Geodynamic | | | Pre-Roman Age |
| | | Geological Time | | | Roman Age |
| | | Fossils and Evolution | | | Middle Age |
| | | Hydrologic Cycle | | | Modern Age |
| | | Landscapes | | Archeology | |
| | | ------------------- | | Anthropology | |
| | Geodynamic | | | Architecture | Rural |
| | Sedimentology | | | | Civil |
| | Stratigraphy | | | | Military |
| | Paleontology | | | | Noble |
| | Structural G. | | | | Cult |
| | Physical Geography | Topography | | | Industrial |
| | | Geomorphic Processes | | Environment | Geosphere |
| | | Geosystems | | | Biosphere |
| | | Climate | | | Biodiversity |
| | | Man – Land interplay | | | BioHazard |
| | | ----------------- | | Folklore | |
| | Geomorphology | | | Museum | Archeological |
| | Hydrogeology | | | | Natural History |
| | Mineralogy | | | | ------------ |
| | Petrography | | | Parks | |
| | Volcanology | | | Educational | Primary School |
| | Geophysics | | | | Secondary School |
| | Seismology | | | | University |
| | Applied G. | | | | Post Graduated |
| | Environmental G. | Geosphere | | | Special needs |
| | | E. Geodiversity | | | CLIL methodology |
| | | E. Geohazard | | | Adult formation |
| | Hazard | Hydrogeological H. | | | Geo-Science divulgation/dissemination |
| | | Seismic H. | | | |
| | | Volcanic H. | | | |
| | GeoHeritage | Territorial GeoHeritage availability, accessibility and usability | | Tourism | Territorial cultural heritage availability, accessibility and usability |
| | | Geodiversity | | | Facilities |
| | | Geo(morpho)sites | | | Attractive |
| | | Geo-Science divulgation/dissemination | | | Folklore |
| | ------------------- | --------------- | | | |

**Figure A1.** Main Geo and non-Geo cultural contexts and themes.

| TIME TARGET Human time scale | | | | | | |
|---|---|---|---|---|---|---|
| MAIN TOPIC - Interplay | TOPICS | | Cultural Context | Theme | Subject | Argument |
| | | Man - Land use Interplay | History | History of man's use of land | Man's land use in no industrialized cultures | Human perception of land and man - land use interplay |
| | | | | | Man's land use in industrialized cultures | Land use and civilization |
| | | | | | | Human perception of man - land use interplay over time |
| | | | Geology | Minerals and rocks | Rocks: genesis and human use | Man/rocks interplay |
| | | | | | | The rule of Rocks in Human Evolution |
| | | | | The Rock Cycle | | |
| | | | Physical Geography | Geomorphic Processes | Exogenous and Endogenous G. P. | Human as Geomorphic Agent |
| | | | | Geosystems | Fluvial systems | |
| | | | | Climate | | Man activities Vs Climate |
| | | | | Landscapes | Timing L. evolution | Man/Landscape interplay |
| | | | Biology | Biodiversity | | Man/ Biodiversity interplay |
| | | Man -Environment Interplay | History | Human perception of environment | Ancient Vs Present environment | Human/Environment interplay |
| | | | Geology | Minerals and rocks | The Rocks Cycle | The rule of Rocks in Environment Evolution |
| | | | | | | |
| | | | Physical Geography | Landscapes | Timing L. evolution | The rule of Landscape in Environment Evolution |
| | | | Biology | Biodiversity | | Environment and biodiversity |
| | | Land – climate Interplay | History | Human perception of climate | Ancient Landscapes | Human/Landscapes interplay |
| | | | Geology | Minerals and rocks | The Rocks Cycle | The rule of Rocks in Landscapes Evolution |
| | | | | | | |
| | | | Physical Geography | Geomorphic Processes | Exogenous G. P. | Human as Geomorphic Agent |
| | | | | | Endogenous G. P. | |
| | | | Biology | Biodiversity | | Climate and Biodiversity |

**Figure A2.** Scheme for Main Topic Interplay and Time Target—Human time scale.

| Cultural context | | |
|---|---|---|
| | **Geology** | **History** |
| **Theme** | *Mineral and Rocks* | *Pre-Roman Age, Mans's use of land,* |
| **Subject** | *Clastic Rocks,* | *Landscape, Climate, Human activities,* |
| **Argument** | *Sedimentary Rocks, Man/Rocks Interplay* | *Man/Rocks Interplay, Fluvial System and Human activities, Human activities and Climate changes.* |

The ancient Greeks believed that everything was made up of four basic elements: *earth*, *water*, *air*, and *fire*. **Geology** is concerning the study of the Earth, *what* it is made of, *why* and *how* it evolves (in the past, present and future time). The Earth is made of rocks and minerals, providing the soil to grow vegetation and support life, and of water. Geologists study land (*earth*), *water* (rivers, oceans and so on), climate, rocks and minerals (*earth*), and how these things change over time, influencing life as we know it now and as it has been in the past. **Rocks** are made of two or more minerals, giving rocks their rough texture. Three families of Rocks are present: *Igneous rocks*, formed when melted rock, called magma or lava, cools and becomes hard; *Sedimentary rocks* formed when small bits (also called sediment) of pre-existing rocks (may be igneous, sedimentary or metamorphic) settle down and stick together, in the sea/lake/river bottom, *Metamorphic rock*s, formed when an old rock (igneous or sedimentary rock) is exposed to a lot of heat and a lot of pressure, this make changes the minerals that the rock is made of and crystals are allowed to grow in the spaces left then in sedimentary rocks. Conglomerate, Sandstone, limestone, and shale are common sedimentary rocks. Sometimes fossils, which are remains or impressions left by plants or animals that got trapped as the rock was forming, are present.

Over time M**an** has always interacted with the environment in which he lived, using what nature put at his disposal. During prehistoric times some rocks were worked and used as tools, precious stones were used to create jewellery; natural cavities were used as a refuge and some rocks were dug to create shelters. Over time, rocks have been used as a building material. Natural cavities, dugged shelters and natural/human made hypogea are present in several ancient sites. Used as a home, as a shelter for animals and foodstuffs, they represent the testimony of the close relationship between man and the environment in which he lives. Human settlements need water for survival, so rivers and springs are fundamental elements to guarantee the continuous human presence in a territory. Over time, the relationship between man and the environment has changed. From being an element belonging to the environment in which man lives, he has become an element that uses the environment and its resources. Man has always used resources such as water and land for its livelihood, for cultivation and breeding. To create areas that can be cultivated or suitable for grazing, it has profoundly acted on the natural environment, modifying it irreversibly, at times. At present, the rapid technological and industrial human progression causes an equally, if not faster, environmental deterioration that affects the entire planet Earth. A more responsible use of resources, such as water, land and air, as well as their protection, is the basis of an ecological and environmentally friendly Man / Land Use approach. As well as responsible behaviors for the protection of the environment are necessary starting from individual citizens up to large industries. Every single citizen can and must do his part for environmental protection.

San Giorgio lucano field trip
San Giorgio Lucano is located on an oriented NE-SW ridge, mainly made of sandstones. Sandstones are sedimentary rocks consisting of grain cemented together (grain size between 0.0625 and 2 mm); larger (grain size more than 2 mm, conglomerates) or smaller (grain size less than 0.0625 mm, clay) grain lenses, as well as fossil remains may be present. At the San Giorgio lucano caves it is possible to observe sandstone, conglomerates and clays outcrops, fossil remains are also present. The presence in the surroundings of archaeological sites, represented by farms, cities and commercial ports, testifies to a territory full of productive activities linked to pastoralism and agricultural cultivation, in the past. The caves complex creation is attributed to the Basilian monks between the VIII and XI centuries, moved from the Byzantine Empire to southern Italy starting from 726 AC, because of Emperor Lion III Isaurico iconoclastic persecution. In southern Italy monks formed several colonies, preferring isolated places, with natural or excavated caves to practise their religion. In San Giorgio lucano the caves were mainly used as cellars/foodstuffs warehouses if exposed to the north or shelter for animals if exposed to the east. Caves, originally, consisted of single or bi-chambers caves; change in time have led to the creation of new rooms. The last room, known as the "grottino", often hosted an altar for the religious rite.

| **Activities** | Talk about experience | Analyze and report on the data collected | | |
|---|---|---|---|---|

**Figure A3.** Descriptive Table Front. Topic: Man—Land use Interplay Educational **Level I**.

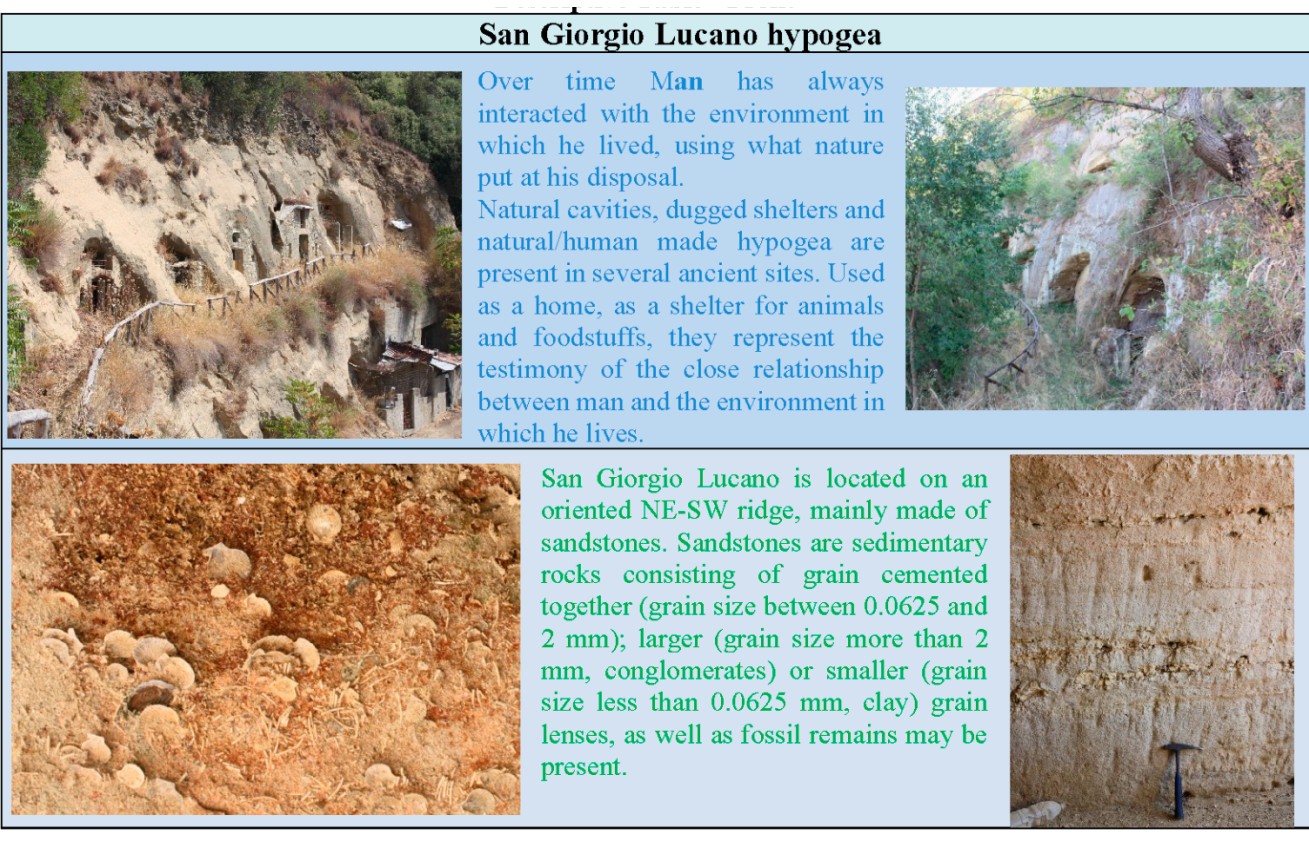

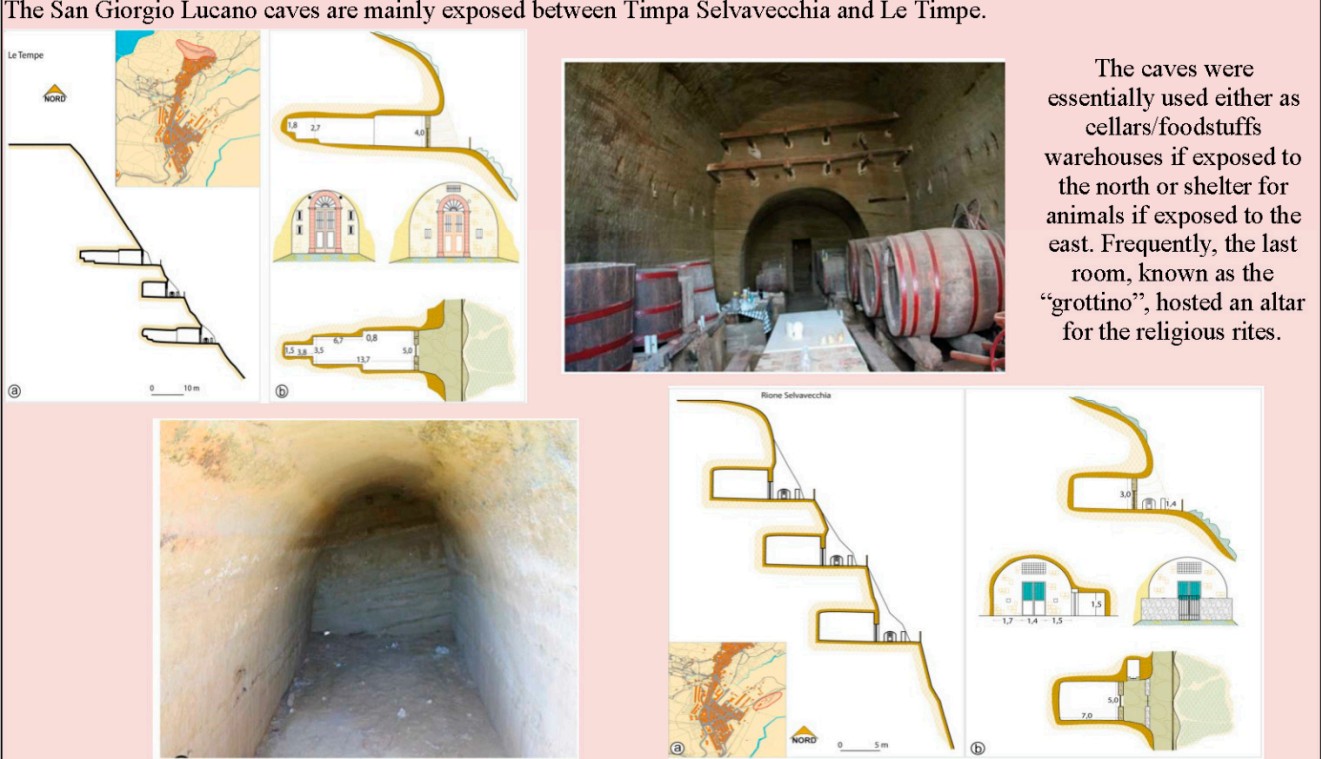

**Figure A4.** Descriptive Table Front. Topic: Man—Land use Interplay Tourism **Level I**.

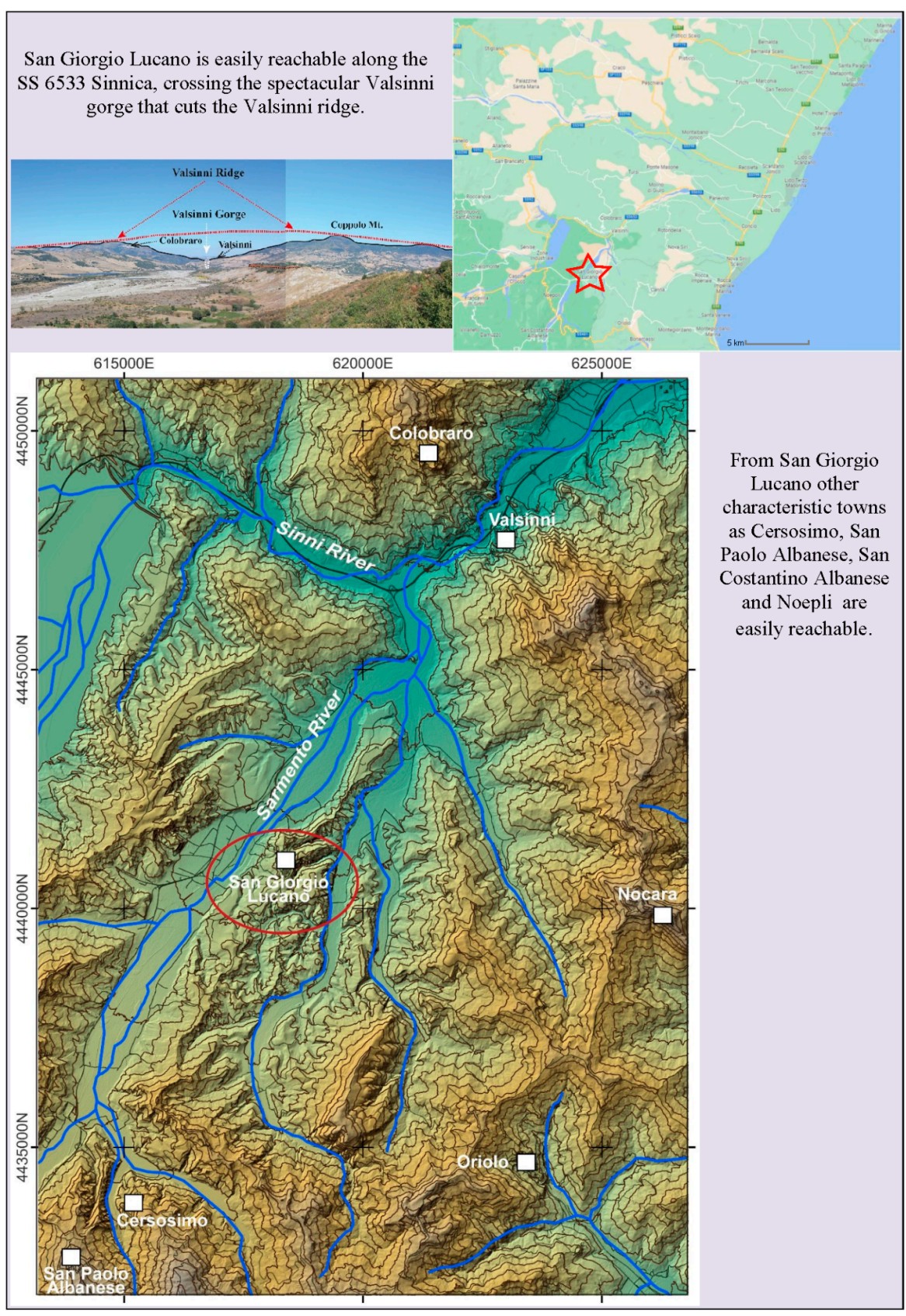

**Figure A5.** Descriptive Table back. Topic: Man—Land use Interplay Tourism **Level I**.

| Physical Geography | | |
|---|---|---|
| **Theme** | *Geomorphic processes* | *Geosystems* | *Climate* |
| **Subject** | *Exogenous forces* | *Fluvial Geosystem* | *Climate changes* |
| **Argument** | *Man as geomorphic agent* | *Dynamic evolution of Fluvial Geosystem Fluvial Geosystem and Human activities* | *Climate changes and Human activities* |

### San Giorgio lucano field trip

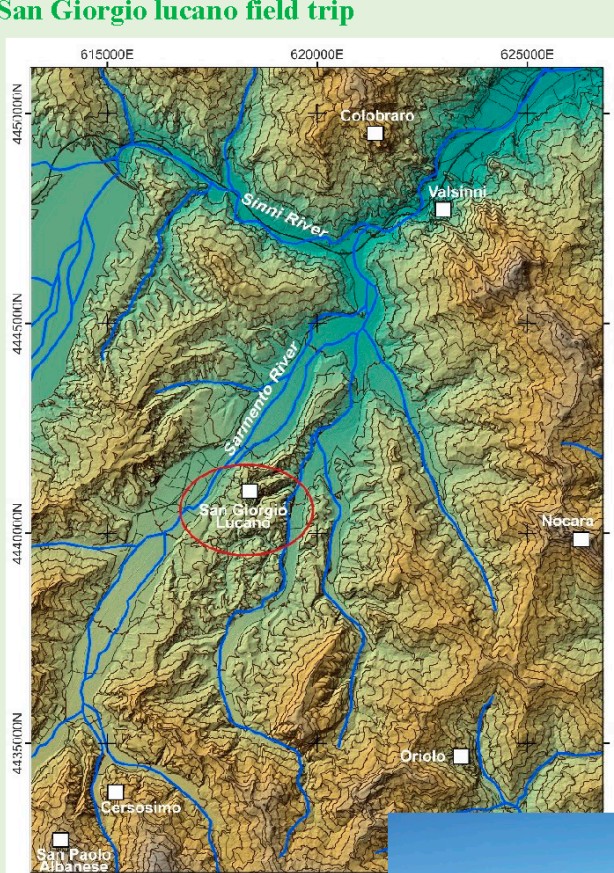

San Giorgio Lucano is located on an oriented NE-SW engraved ridge, on the Sarmento river's hydrographic right, before it flows into the Sinni river. The ridge is bordered towards NW by the Sarmento river and towards SE by the Fiumarella stream; in correspondence of T.mpa Ciucca (669 m) the ridge changes direction towards SE and the Lapio stream separates it from NW-SE oriented ridges on which the towns of Cersosimo and San Paolo Albanese are present. The ridge is mainly made by sandstones and shales, conglomerates are also present

Many landforms are featured in the physical landscape of the Valsinni area.

The present-day floodplain of the Sinni River is the main feature (number 0) and is next to three older fluvial terraces indicated as number 1, 2, and 3 which are few meters higher than the floodplain. All these landforms represent the flat area of the Sinni River floor valley. The highest fluvial terrace shown on the right side valley testify a past base level much higher than the one. In the same valley flank badland slope are also present indicated by an absence of vegetation.

In the background the large V-shaped valley cuts by the Sinni River (Valsinni gorge) is evidenced

Red dashed line represents the enveloped line of the mountain peaks in the V-shaped Valsinni valley and can suggests what has been the vertical incision and what have been the eroded surface starting from the top-surface watershed of the Valsinni Ridge.

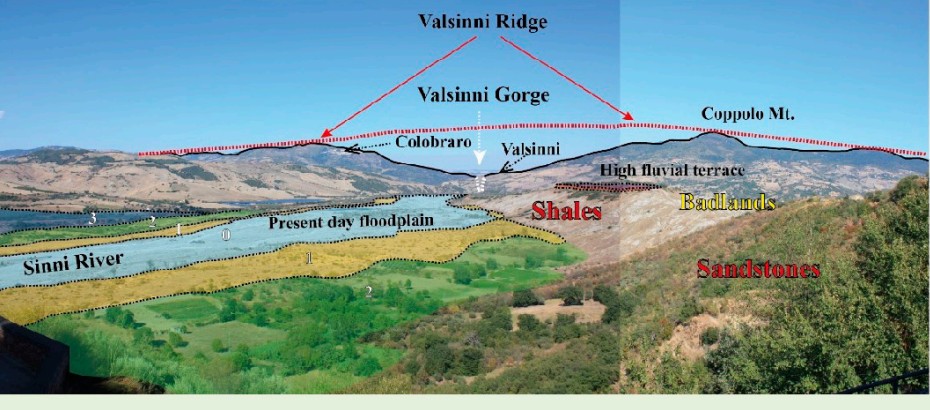

Over time this area has experienced periods of intense agricultural and pastoral use and periods of abandonment and depopulation.

| Activities | Reports Cartographic activities Research and comparison between similar geomorphic contexts Use of specific GIS's tools |
|---|---|

**Figure A6.** Descriptive Table. Topics: Man—Land use Interplay, Man—Environment Interplay, and Land—Climate Interplay. Educational **Level III**.

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
