# Peer review of "Geoheritage Management in Areas with Multicultural Interest Contexts"

_sustainability, doi:10.3390/su142315911_

Round 1

Reviewer 1 Report

Review Report- Ananda Y. Karunarathne/PhD

This manuscript analyzed Geoheritage management in areas with multicultural interest contexts.  

The scope of this work is quite significant, but the manuscript did not sufficiently explain the empirical data collection under the research methods, and implications of the research findings. Having looked the paper over carefully, specifically, I have following major concerns and suggestions to the authors, which are listed below: 

1). In the Abstract, authors should explain concisely, about the methodology that they occupied/used for their study. Authors should also be more specific about the method/s they used. 

2). In general, this manuscript is written in such a manner as both the details underlying the research motivation and the research methodology are not clearly articulated. To help readers better understand the key method/s, I strongly recommend authors should explain vividly the base mechanism that they used/occupied for preparing informative and territorial promotion material, in the methodology section. To help readers better understand the methodology, I would also suggest authors to add a flowchart to portray the methodological approach of the study in the methodology section, concisely.

3). Regarding explaining the Man-environment nexus, I would suggest authors also to touch the theories of Environmental Determinism and Possibilism, may be in the introduction section.

4).  I can’t see the research question/s of this study. I kindly suggest authors explain concisely the research question/s of this work, may be in the introduction part.

5). The Data Collection procedure of the study is not clear. What are the data collection tool/s that authors used to select a portion of the Sinni river's catchment area (Basilicata region, Southern Italy). Seemingly, the authors are not clear about their data collection mechanism. I strongly recommend authors need to explain mainly what are their data? Are they primary or secondary?, and what is their data collection mechanism? (basically, I found that some authors have responsible for data curation, this is because readers need overall picture about the data collection/field observation process).

6). Authors should explain, what are the criteria that they used for selecting the study sites.

7). The title of the section 2 needs to be changed. I would suggest authors to keep just “Methods” and they can explain “Aims” of the study in the Introduction section. Secondly regarding the section 5, I would suggest using “Conclusion or Concluding remarks” instead of final remarks. 

8). I cannot see how this manuscript contributes much to the existing body of literature on the topic, even in an applied context. Authors need to explain clearly, what is the intellectual merit or innovative application of this work? I suggest authors to add this in the introduction section.

9). I kindly suggest the authors to separate the discussion section from the results section.

10). A discussion of the weaknesses and limitations of the method/study used is missing which could guide fellow researchers. Authors can concisely discuss this adding two or three sentences concisely in the conclusion section.

11). There are some typological errors throughout the manuscript. I would suggest the authors to carefully go through the manuscript and correct them.  

12). (i). I also found some contradictions about followings;

“….how the geographical-physical aspects are decisive for the territorial area characterization…”(line 150-151, page 4).

I would suggest using “physical geographical aspects”, instead of geographical-physical aspects.

(ii). Authors should put/cite the sources/references of key figures (e.g. figure 1 to figure 4; Fig. 6, Fig. 11 etc)?.  

(iii). Regarding the “Scheme of cultural context”/Figure 9. I found a controversy, how authors put the “Environment” under the Non Geo category?

Good Luck!!!

Author Response

Review 1 Ananda Y. Karunarathne

 Referee comment: 1) In the Abstract, authors should explain concisely, about the methodology that they occupied/used for their study. Authors should also be more specific about the method/s they used.

Reply: Thank you for your suggestion. Some sentences have been added in both the abstract and the text explaining better the method used in the paper. The method is also shown in the new figure 3. As an example the following text has been added in the Abstract “…This paper represents results of a qualitative study providing an overview of the possibility, in a multicultural context, about whether, when and how the geo context may act as a link between the different disciplines, and what is the best way to do so…”

Referee comment: 2). In general, this manuscript is written in such a manner as both the details underlying the research motivation and the research methodology are not clearly articulated. To help readers better understand the key method/s, I strongly recommend authors should explain vividly the base mechanism that they used/occupied for preparing informative and territorial promotion material, in the methodology section. To help readers better understand the methodology, I would also suggest authors to add a flowchart to portray the methodological approach of the study in the methodology section, concisely.

Reply: Thank you for your suggestion. A better explanation of the mechanisms used for preparing informative and territorial promotion has been done. The following texts have been added in the Methods and Aim section. “Figure 3 shows the procedure followed in the study. Through successive steps, arranged into three stages and characterized by Actions (What) on Objects (Who) based on Criteria / Methods (How), final materials (i.e. Tables, Descriptive Tables, Correlation schemes, etc.), prepared according to the end user needs, have been produced.

Referee comment: 3). Regarding explaining the Man-environment nexus, I would suggest authors also to touch the theories of Environmental Determinism and Possibilism, may be in the introduction section.

Reply: It’s an interesting suggestion, thank you. Unfortunately, these theories are included in the context of Human Geography and are not considered in this paper. In the future we will also add this context in our papers; currently, we consider that the time granted for required text changes is not enough to provide a correct framework, even if only hinted at.

Referee comment: 4). I can’t see the research question/s of this study. I kindly suggest authors explain concisely the research question/s of this work, may be in the introduction part.

Reply: Thank you for your suggestion. The research question has been added in the Abstract and developed in The Introduction section.

Referee comment: 5). The Data Collection procedure of the study is not clear. What are the data collection tool/s that authors used to select a portion of the Sinni river's catchment area (Basilicata region, Southern Italy). Seemingly, the authors are not clear about their data collection mechanism. I strongly recommend authors need to explain mainly what are their data? Are they primary or secondary?, and what is their data collection mechanism? (basically, I found that some authors have responsible for data curation, this is because readers need an overall picture of the data collection/field observation process).

Reply: Thank you for your suggestion. Figure 3, and related text which has been added in the Methods and Aim section provides the information you requested.

Referee comment: 6). Authors should explain, what are the criteria that they used for selecting the study sites.

Reply: Please, see the above comment.

Referee comment: 7). The title of the section 2 needs to be changed. I would suggest authors to keep just “Methods” and they can explain “Aims” of the study in the Introduction section. Secondly regarding the section 5, I would suggest using “Conclusion or Concluding remarks” instead of final remarks.

Reply: Thank you for your observation. Aims are concisely illustrated in the Introduction section and are subsequently explained in a more extensive form in the Methods ad Aims section. We change the section “Final remarks” as “Concluding remarks”.

Referee comment: 8). I cannot see how this manuscript contributes much to the existing body of literature on the topic, even in an applied context. Authors need to explain clearly, what is the intellectual merit or innovative application of this work? I suggest authors to add this in the introduction section.

Reply: The innovation of this application has been explained in the following text which has been added in the Introduction section. “Most of the literature on heritage issues is specific and sectoral, dedicated to a single cultural context (historical, archaeological, geological, etc.). In a multicultural context, discriminating whether, when and how the geo-context can act as a link between different disciplines, and what is the best way to do so, is a further way of addressing heritage issues. Geo aspects may represent a topic around which, and with which, prepare materials and contents for educational, informative and tourist purposes. This multidisciplinary approach can also be used for the production of materials and contents to support territorial planning actions, aimed at territorial enhancing and protecting. In this paper, just aspects related to educational, informative, and touristic purposes are considered”.

Referee comment: 9). I kindly suggest the authors to separate the discussion section from the results section.

Reply: We are sorry, but we do not agree and prefer to maintain the current organization of the text.

Referee comment: 10). A discussion of the weaknesses and limitations of the method/study used is missing which could guide fellow researchers. Authors can concisely discuss this adding two or three sentences concisely in the conclusion section.

Reply: Following your suggestion, the new sentences have been added in the Concluding remarks section: “The main critical issues encountered in writing this paper concern the difficulty of finding the necessary material, the different criteria followed in the processing of data and the use of key words. In particular: papers are not always easily available online; the approach to data (in terms of collection, cataloging, processing, and restitution) varies according to the cultural context considered; sometimes key words are used with different meanings depending on the cultural context taken into account”.

Referee comment: 11). There are some typological errors throughout the manuscript. I would suggest the authors to carefully go through the manuscript and correct them.

Reply: done.

Referee comment: 12).

(i). I also found some contradictions about followings; “….how the geographical-physical aspects are decisive for the territorial area characterization…”(line 150-151, page 4). I would suggest using “physical geographical aspects”, instead of geographical-physical aspects.

Reply: We apologize for the refuse, the text has been modified.

(ii). Authors should put/cite the sources/references of key figures (e.g. figure 1 to figure 4; Fig. 6, Fig. 11 etc)?.

Reply: Figures from 1 to 4, figure 6, and figure from 9 to 11, are original and unpublished.

(iii). Regarding the “Scheme of cultural context”/Figure 9. I found a controversy, how authors put the “Environment” under the Non Geo category?

Reply: We apologize for this misunderstanding, we consider Environmental Geology in the Geo category. The environment in the Non Geo category is referred to as the Environment biological aspect. According to your comment, we modified the figure caption as follows: “Figure 9. Scheme of cultural context showing the hierarchization and correlation of Theme, Subject, and Arguments related to History, Geology, and Physical Geography. The environment in Non Geo context is referred to as its biological matrices. In red Themes, Subjects, and Arguments considered for the analyzed area are highlighted”.

Reviewer 2 Report

This is a very intresting article. However, the document requires extensive English language editing to make it more readable. As is the document is extremely difficult to read.

Author Response

Referee comment: This is a very intresting article. However, the document requires extensive English language editing to make it more readable. As is the document is extremely difficult to read.

Reply: Thank you for your suggestion. The English text has been revised by a native English speaker.

Reviewer 3 Report

Dear authors, thanks for a well done study. Please enhance the implications of the study by elaborating the practical and policy implications of the study. In two or three paragraphs explain who will benefit from the study and how.  

Author Response

Referee comment: Dear authors, thanks for a well done study. Please enhance the implications of the study by elaborating the practical and policy implications of the study. In two or three paragraphs explain who will benefit from the study and how.

Reply: Thank you for your suggestion. We are currently preparing a further paper dedicated to the management of cultural heritage (geo and non geo), in which we will address the issue of practical and policy implications, which represent a topic that is beyond the context of this paper. The following text has been added in the Introduction section. “Most of the literature on heritage issues is specific and sectoral, dedicated to a single cultural context (historical, archaeological, geological, etc.). In a multicultural context, discriminating whether, when and how the geo-context can act as a link between different disciplines, and what is the best way to do so, is a further way of addressing heritage issues. Geo aspects may represent a topic around which, and with which, prepare materials and contents for educational, informative, and touristic purposes. This multidisciplinary approach can also be used for the production of materials and contents to support territorial planning actions, aimed at territorial enhancing and protecting. In this paper, just aspects related to educational, informative and tourist purposes are considered. So, the main end-users considered are represented by primary and secondary school students, university students, and tourists, for whom the contents have been prepared (concepts and vocabulary) according to the corresponding level considered”.

Reviewer 4 Report

Congratulations to the Authors.

The study analyses the geoheritage sites as a basis for the upgrading/development of sites of multifaceted cultural and natural value, which benefits the development and management of the areas

The topic has its background, and is a topical subject of discussion today, alongside the increase in publications in the direction of geoheritage. The research can be defined as relevant to the field.

It cannot be claimed that the study fills information/thematic gaps, but it thoroughly interprets the topic of the practical use of diverse territorial information of public interest (heritage, culture, education), which is directly relevant to the current topic of integrated territorial management with the active participation of stakeholders.

The text pays underscored attention to the theoretical setting, which is much needed given the subject matter. For me, this text is complete. Further tests of the methodology and practical examples are needed, which the authors indicate are forthcoming.

The conclusions are consistent with the evidence and arguments presented and the author's style of exposition is excellent and the logical connection in the text is clearly followed. This also applies to the content of the conclusions.

The references are appropriate. I take them to be sufficient in terms of the emphases placed in the text and the important terminological definitions derived.

An impressive study in terms of its depth and multi-faceted nature, with high application value both in terms of educational needs and territorial management. I recommend the authors carefully review the cartographic material - not everywhere the legends are readable. The tables could also be lightened in terms of color loading.

Author Response

Referee comment: It cannot be claimed that the study fills information/thematic gaps, but it thoroughly interprets the topic of the practical use of diverse territorial information of public interest (heritage, culture, education), which is directly relevant to the current topic of integrated territorial management with the active participation of stakeholders.

The text pays underscored attention to the theoretical setting, which is much needed given the subject matter. For me, this text is complete. Further tests of the methodology and practical examples are needed, which the authors indicate are forthcoming.

The conclusions are consistent with the evidence and arguments presented and the author's style of exposition is excellent and the logical connection in the text is clearly followed. This also applies to the content of the conclusions. The references are appropriate. I take them to be sufficient in terms of the emphases placed in the text and the important terminological definitions derived.

An impressive study in terms of its depth and multi-faceted nature, with high application value both in terms of educational needs and territorial management. I recommend the authors carefully review the cartographic material - not everywhere the legends are readable. The tables could also be lightened in terms of color loading.

Reply: Thank you for your suggestion and as you requested, details on the methodology have been added. Furthermore, a review of the cartographic material has been done to produce more readable legends enlarging the frame area.

Round 2

Reviewer 1 Report

Authors have sufficiently improved the manuscript. 

Good Luck!!!